# COUNTERFACTUALS UNCOVER THE MODULAR STRUCTURE OF DEEP GENERATIVE MODELS

**Michel Besserve**[1,2,*]**, Arash Mehrjou**[1,3]**, Rémy Sun**[1,3]**, Bernhard Schölkopf**[1]

1. MPI for Intelligent Systems, Tübingen, Germany.
2. MPI for Biological Cybernetics, Tübingen, Germany.
3. Department of Computer Science, ETH Zürich, Switzerland.
4. ENS Rennes, France.

## ABSTRACT

Deep generative models can emulate the perceptual properties of complex image datasets, providing a latent representation of the data. However, manipulating such representation to perform meaningful and controllable transformations in the data space remains challenging without some form of supervision. While previous work has focused on exploiting statistical independence to *disentangle* latent factors, we argue that such requirement can be advantageously relaxed and propose instead a non-statistical framework that relies on identifying a modular organization of the network, based on counterfactual manipulations. Our experiments support that modularity between groups of channels is achieved to a certain degree on a variety of generative models. This allowed the design of targeted interventions on complex image datasets, opening the way to applications such as computationally efficient style transfer and the automated assessment of robustness to contextual changes in pattern recognition systems.

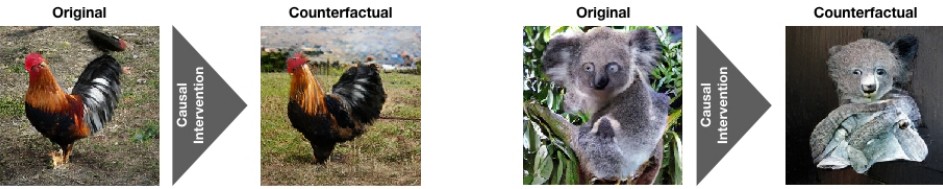

Figure 1: Counterfactual manipulation of samples of a BigGan trained on ImageNet (see section 4.2).

## 1 INTRODUCTION

Deep generative models, by learning a non-linear function mapping a latent space to a space of observations, have proven successful at designing realistic images in a variety of complex domains (objects, animals, human faces, interior scenes). In particular, two kinds of approaches emerged as state-of-the-art (SOTA): Generative Adversarial Networks (GAN) (Goodfellow et al., 2014), and Variational Autoencoders (VAE) (Kingma & Welling, 2013; Rezende et al., 2014).

Efforts have been made to have such models produce *disentangled* latent representations that can control interpretable properties of images (Kulkarni et al., 2015; Higgins et al., 2017). However, the resulting models are not necessarily *mechanistic* (or *causal*) in the sense that interpretable properties of an image cannot be ascribed to a particular part, a *module*, of the network architecture. Gaining access to a modular organization of generative models would benefit the *interpretability* and allow *extrapolations*, such as generating an object in a background that was not previously associated with this object, as illustrated in a preview of our experimental results in Fig. 1.

Such extrapolations are an integral part of human representational capabilities (consider common expressions such as "like an elephant in a china shop") and consistent with the modular organization

---

*michel.besserve@tuebingen.mpg.de

of its visual system, comprising specialized regions encoding objects, faces and places (see e.g. Grill-Spector & Malach (2004)). Extrapolations moreover likely support adaptability to environmental changes and robust decision making (Dvornik et al., 2018). How to leverage trained deep generative architectures to perform such extrapolations is an open problem, largely due to the non-linearities and high dimensionality that prevent interpretability of computations performed in successive layers.

In this paper, we propose a causal framework to explore modularity, which relates to the causal principle of *Independent Mechanisms*, stating that the causal mechanisms contributing to the overall generating process do not influence nor inform each other (Peters et al., 2017).[1] We study the effect of direct interventions in the network from the point of view that the mechanisms involved in generating data can be modified individually without affecting each other. This principle can be applied to generative models to assess how well they capture a causal mechanism (Besserve et al., 2018a). Causality allows to assay how an outcome would have changed, had some variables taken different values, referred to as a *c*ounterfactual (Pearl, 2009; Imbens & Rubin, 2015). We use counterfactuals to assess the role of specific internal variables in the overall functioning of trained deep generative models, along with a rigorous definition of disentanglement in a causal framework. Then, we analyze this disentanglement in implemented models based on unsupervised counterfactual manipulations. We show empirically how VAEs and GANs trained on image databases exhibit modularity of their hidden units, encoding different features and allowing counterfactual editing of generated images.

**Related work.** Our work relates to the interpretability of convolutional neural networks, which has been intensively investigated in discriminative architectures (Zeiler & Fergus, 2014; Dosovitskiy & Brox, 2016; Fong & Vedaldi, 2017; Zhang et al., 2017b;a). Generative models require a different approach, as the downstream effect of changes in intermediate representations are high dimensional. InfoGANs. $\beta$-VAEs and other works (Chen et al., 2016; Mathieu et al., 2016; Kulkarni et al., 2015; Higgins et al., 2017) address supervised or unsupervised disentanglement of latent variables related to what we formalize as *extrinsic disentanglement* of transformations acting on data points. We introduce the novel concept of *intrinsic disentanglement* to uncover the internal organization of networks, arguing that many interesting transformations are statistically dependent and are thus unlikely to be disentangled in the latent space. This relates to Bau et al. (2018) who proposed a framework based on interventions on internal variables of a GAN which, in contrast to our fully unsupervised approach, requires semantic information. Higgins et al. (2018) suggest a definition of disentanglement based on group representation theory. Compared to this proposal, our approach (introduced independently in (Besserve et al., 2018b)) is more flexible as it applies to arbitrary continuous transformations, free from the strong requirements of representation theory (see Appendix F). Finally, an interventional approach to disentanglement has also be taken by Suter et al. (2018), who focuses on extrinsic disentanglement in a classical graphical model setting and develop measures of interventional robustness based on labeled data.

## 2 FROM DISENTANGLEMENT TO COUNTERFACTUALS AND BACK

We introduce a general framework to formulate precisely the notion of disentanglement and bridge it to causal concepts. This theory section will be presented informally to ease high level understanding. Readers interested in the mathematical aspects can refer to Appendix A where we provide all details.

### 2.1 A CAUSAL GENERATIVE MODEL (CGM) FRAMEWORK

We consider a generative model $M$ that implements a function $g_M$, which maps a latent space $\mathcal{Z}$ to a manifold $\mathcal{Y}_M$ where the learned data points live, embedded in ambient Euclidean space $\mathcal{Y}$ (Fig. 2a). A sample from the model is generated by drawing a realization $z$ from a prior latent variable distribution with mutually independent components, fully supported in $\mathcal{Z}$. We will use the term *representation* to designate a mapping $r$ from $\mathcal{Y}_M$ to some representation space $\mathcal{R}$ (we also call $r(y)$ the representation of a point $y \in \mathcal{Y}_M$). In particular, we will assume (see Def. 7 and Prop. 4, in Appendix A) that $g_M$ is (left)-invertible, such that $g_M^{-1}$ is a representation of the data, called the *latent representation*.

Assuming the generative model is implemented by a non-recurrent neural network, we can use a causal graphical model representation of the entailed computational graph implementing the mapping $g_M$ through a succession of operations (called *functional assignments* in causal language), as illustrated

---

[1]Note that this is not a *statistical* independence; the quantities transformed by the mechanisms of course do influence each other and can be statistically dependent.

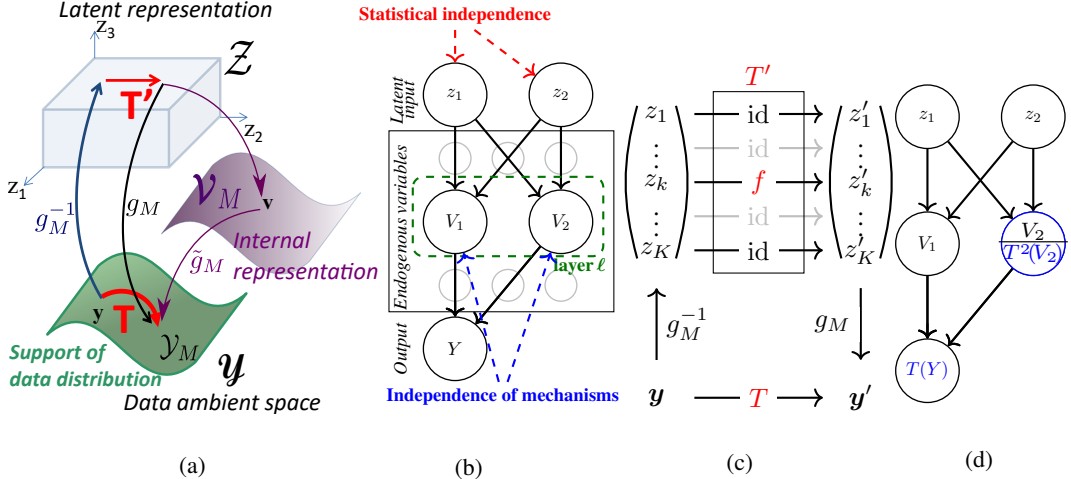

Figure 2: (a) Illustration of the generative mapping and a disentangled transformation. (b) Causal graph of an example CGM showing different types of independence between nodes. (c) Commutative diagram showing sparse transformation $T'$ in latent space associated to a disentangled transformation $T$. (d) Illustration of intrinsic disentanglement with $\mathcal{E} = \{2\}$.

in Fig. 2b, that we will call Causal Generative Model (CGM). In addition to the latent representation, we can then choose a collection of possibly multi-dimensional endogenous (internal) variables represented by nodes in the causal graph, such that the mapping $g_M$ is computed by composing the *endogenous variable assignment* $v_M$ with the *endogenous mapping* $\tilde{g}_M$ according to the diagram

$$ \mathcal{Z} \overset{v_M}{\to} \boldsymbol{\mathcal{V}}_M \overset{\tilde{g}_M}{\to} \mathcal{Y}_M \,. $$

A paradigmatic choice for these variables is the collection of output activation maps of each channel in one hidden layer of a convolutional neural network, as illustrated in Fig. 2b. As for the latent case, we use mild conditions that guarantee $\tilde{g}_M$ to be left-invertible, defining the *internal representation* of the network (see Def. 7 and Prop. 4 in Appendix A). Given the typical choice of dimensions for latent and endogenous variables, the $V_k$'s are also constrained to take values in subsets $\mathcal{V}_M^k$ of smaller dimension than their Euclidean ambient space $\mathcal{V}^k$. As detailed in Appendix A, we introduce the *endogenous image* sets of the form $\boldsymbol{\mathcal{V}}_M^{\mathcal{E}} = \big\{ \boldsymbol{v} \in \prod_{k \in \mathcal{E}} \mathcal{V}^k \colon \boldsymbol{v} = (v_k(\boldsymbol{z}))_{k \in \mathcal{E}}, \, \boldsymbol{z} \in \mathcal{Z} \big\}$ for a subset of variables indexed by $\mathcal{E}$ (amounting to $\boldsymbol{\mathcal{V}}_M$ when $\mathcal{E}$ includes all endogenous variables).

This CGM framework allows defining counterfactuals in the network following Pearl (2014).

**Definition 1** (Unit level counterfactual, informal). *Given CGM $M$, the* interventional model $M_{\boldsymbol{h}}$ *is obtained by replacing assignments of the subset variables $\mathcal{E}$ and by the vector of assignments $\boldsymbol{h}$. Then for any latent input $\boldsymbol{z}$, called* unit, *the* unit-level counterfactual *is the output of $M_{\boldsymbol{h}}$:*

$$ Y_{\boldsymbol{h}}^{\mathcal{E}} = g_{M_{\boldsymbol{h}}}(\boldsymbol{z}) \,. $$

Def. 1 is also in line with the concept of *potential outcome* (Imbens & Rubin, 2015). Importantly, counterfactuals induce a transformation of the output of the generative model.

**Definition 2** (Counterfactual mapping). *Given an embedded CGM, we call the transformation*

$$ \overset{\frown}{Y}{}_{\boldsymbol{h}}^{\mathcal{E}} : y \mapsto g_{M_{\boldsymbol{h}}} \big( g_M^{-1}(y) \big) $$

*the $\boldsymbol{h}$-counterfactual mapping. We say it is* faithful *to $M$ whenever $\overset{\frown}{Y}{}_{\boldsymbol{h}}^{\mathcal{E}} [\mathcal{Y}_M] \subset \mathcal{Y}_M$.*

We introduce faithfulness of a counterfactual mapping to account for the fact that not all interventions on internal variables will result in an output that could have been generated by the original model. In the context of generative model, non-faithful counterfactuals generate examples that leave the support of the distribution learned from data, possibly resulting in an artifactual output (assigning a large value to a neuron may saturate downstream neurons), or allowing extrapolation to unseen data.

## 2.2 UNSUPERVISED DISENTANGLEMENT: FROM STATISTICAL TO CAUSAL PRINCIPLES

The classical notion of disentangled representation (e.g. Bengio et al. (2013); Kulkarni et al. (2015)), posits individual latent variables "*sparsely encode real-world transformations*". Although the concept of *real-world transformations* remains elusive, this insight, agnostic to statistical concepts, has driven supervised approaches to disentangling representations, where relevant transformations are well-identified and manipulated explicitly using appropriate datasets and training procedures.

In contrast, unsupervised learning approaches to disentanglement need to learn such *real-world transformations* from unlabeled data. In order to address this challenge, SOTA approaches seek to encode such transformations by changes in individual latent factors, and resort to a statistical notion of disentanglement, enforcing conditional independence between latent factors (Higgins et al., 2017). This statistical approach leads to several issues:

- **Independence is not necessary.** The i.i.d. constraints on the latent variables' prior impose statistical independence between disentangled factors for the data distribution. This is unlikely for many relevant properties, counfounded by factors of the true data generating mechanisms (e.g. skin and hair color).
- **Independence is not sufficient** to specify a disentangled representation, such that the problem remains ill-posed (Locatello et al., 2018). As a consequence, finding an appropriate inductive bias to learn a representation that benefits downstream tasks remains an open question.
- To date, SOTA unsupervised approaches are mostly demonstrated on synthetic datasets, and beyond MNIST **disentangling complex real world data has been limited** to the well-calibrated CelebA dataset. On complex real-world datasets, **disentangled generative models exhibit visual sample quality far below non-disentangled SOTA** (e.g. BigGAN exploited in our work (Brock et al., 2018)).

We propose an non-statistical definition of disentanglement by first phrasing mathematically the transformation-based insights (Bengio et al., 2013; Kulkarni et al., 2015). Consider a transformation $T$ acting on the data manifold $\mathcal{Y}_M$. As illustrated by the commutative diagram of Fig. 2c, disentanglement of such property then amounts to having $T$ correspond to a transformation $T'$ of the latent space that would act only on a single variable $z_k$, using transformation $f$, leaving the other latent variables available to encode other properties. More explicitly we have

$$T'(\boldsymbol{z}) : (z_1, .., z_k, .., z_K) \mapsto (z_1, .., f(z_k), .., z_K), \quad \text{and} \quad T(g_M(\boldsymbol{z})) = g_M(T'(\boldsymbol{z})).$$

It is then natural to qualify two transformations $T_1$ and $T_2$ as disentangled (from each other), whenever they modify different components of the latent representation (see Def. 10 in Appendix F). This amounts to saying that the transformations follow the causal principle of independent mechanisms (Peters et al., 2017; Parascandolo et al., 2018).

Due to the fact that it relies on transformations of the latent representation, that are exogenous to the CGM, we call this notion *extrinsic disentanglement*. This "functional" definition has the benefit of being agnostic the the subjective choice of the property to disentangle, and to the statistical notion of independence. However, we can readily notice that, if applied to the latent space (where components are i.i.d. distributed), this functional notion of disentangled transformation still entails statistical independence between disentangled factors. We thus need to exploit a different representation to uncover possibly statistically related properties, but disentangled in the sense of our definition.

## 2.3 DISENTANGLING BY MANIPULATING INTERNAL REPRESENTATIONS

As illustrated in the CGM of Fig. 2b, in contrast to latent variables, properties encoded by endogenous variables of the graphical model are not necessarily statistically independent due to common latent causes ($z_1$ and $z_2$ in the figure), but may still reflect interesting properties of the data that can be intervened on independently, following the principle of independence of mechanisms. We thus extend our definition of disentanglement to allow transformations of the internal variables of the network as follows.

**Definition 3** (Intrinsic disentanglement, informal). *In a CGM $M$, a transformation $T : \mathcal{Y}_M \rightarrow \mathcal{Y}_M$ is intrinsically disentangled with respect to a subset $\mathcal{E}$ of endogenous variables, if there is a transformation $T'$ acting on the internal representation space such that for any endogenous value $\boldsymbol{v}$*

$$T(\widetilde{g}_M(\boldsymbol{v})) = \widetilde{g}_M(T'(\boldsymbol{v})) \tag{1}$$

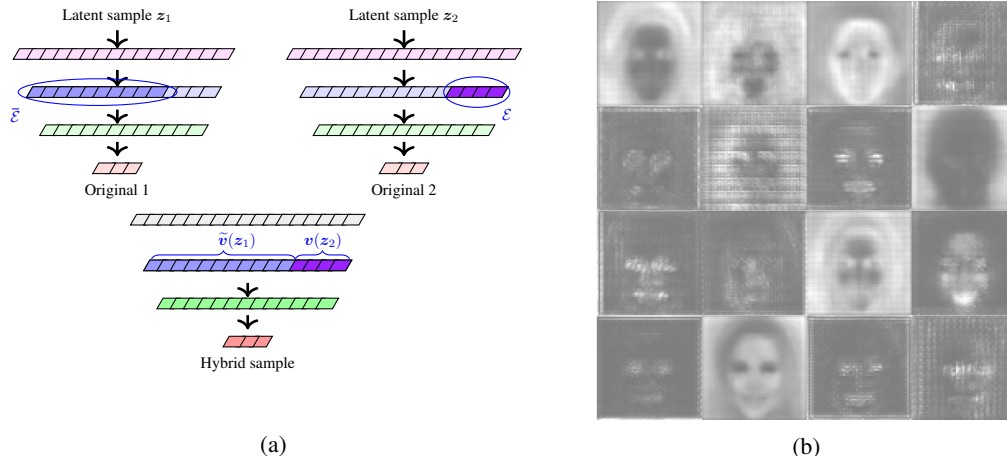

(a)                                                    (b)

Figure 3: **Generation of influence maps.** (a) Principle of sample hybridization through counterfactuals. (b) Examples of EIM generated by a VAE on the CelebA dataset (lighter pixel indicate larger variance and thus stronger influence of the perturbations on that pixel).

*where $T'(\boldsymbol{v})$ only affects the variables indexed by $\mathcal{E}$.*

Fig. 2d illustrates this second notion of disentanglement, where the split node indicates that the value of $V_2$ is computed as in the original CGM (Fig. 2b) before applying transformation $T^2$ to the outcome. While the above definition applies to a single transformation, straightforward extensions of this concept to families of transformations are provided in Appendix F. Faithful counterfactuals represent examples of disentangled transformations:

**Proposition 1** (Counterfactuals and disentanglement, informal)**.** *Consider an intervention on subset $\mathcal{E}$, its associated counterfactual mapping is faithful if and only if it is disentangled. For interventions on variables that remain within the support of the original marginal distribution, it is sufficient that $\mathcal{E}$ and its complement $\bar{\mathcal{E}}$ do not have common latent ancestors.*

This indicates that finding faithful counterfactuals can be used to learn disentangled transformations.

## 3 Finding modularity in deep generative models

### 3.1 Defining modularity

Building on Sec. 2, we define modularity as a structural property of the internal representation, allowing (with the immediately following Prop. 2) to implement arbitrary disentangled transformations.

**Definition 4** (Modularity)**.** *A subset of endogenous variables $\mathcal{E}$ is called modular whenever $\mathcal{V}_M$ is the Cartesian product of $\mathcal{V}_M^{\mathcal{E}}$ by $\mathcal{V}_M^{\bar{\mathcal{E}}}$.*

**Proposition 2** (Modularity implies disentanglement)**.** *If $\mathcal{E}$ is modular, then any transformation applied to it staying within its input domain is disentangled.*

The proof is a natural extension of the proof of Proposition 1. Both the Definition and the Proposition have trivial extensions to multiple modules (along the line described in Appendix F). While we have founded this framework on a functional definition of disentanglement that applies to transformations, the link made here with an structural property of the trained network allows us to define a *disentangled representation* as follows.

**Definition 5** (Disentangled representation)**.** *A disentangled representation is a partition of the endogenous variables in several modules, such that $\mathcal{V}_M$ is the Cartesian product of the corresponding endogenous image sets.*

Then any transformation applied to a given module leads to a valid transformation in the data space (it is relatively disentangled following Def. 10 in Appendix F).

Interestingly, a disentangled representation is associated to a partition of the considered set of endogenous variables into modules. This extra requirement was not considered in classical approaches to disentanglement as it was assumed that each single scalar variables could be considered as an independent module. Our framework provides an insight relevant to artificial and biological systems: as the activity of multiple neurons can be strongly tied together, the concept of representation may not be meaningful at the "atomic" level of single neurons, but require to group them into modules forming a "mesoscopic" level, at which each group can be intervened on independently.

## 3.2 HYBRIDIZATION AS A DISENTANGLED TRANSFORMATION

As stated in Sec. 2, a functional definition of disentanglement, leaves unanswered how to find relevant transformations. Prop. 1 and 2 provide the following hints: (1) Once a modular structure is found in the network, a broad class of disentangled transformations are available, (2) Transformations that stay within their input domain are good candidates of disentanglement, (3) Counterfactual interventions implicitly define transformations. We follow these guidelines by assigning a constant value $v_0$ to a subset of endogenous variables $\mathcal{E}$ to define counterfactuals (i.e. $h$ is a constant function), aiming for faithful ones by constraining $v_0$ to belong to $\mathcal{V}_M^{\mathcal{E}}$. To avoid characterizing $\mathcal{V}_M^{\mathcal{E}}$, we rely on sampling from the (joint) marginal distribution of the variables in $\mathcal{E}$.

To illustrate the procedure, we consider a standard feed-forward multilayer neural network and choose endogenous variables to be the collection of all output activations of channels of a given layer $\ell$. Let $\mathcal{E}$ be a subset of these channels, the hybridization procedure, illustrated in Fig. 3a goes as follows. We take two independent examples of the latent variable $z_1$ and $z_2$, that will generate two *original* examples of the output $(y_1, y_2) = (g_M(z_1), g_M(z_2))$ (that we call *Original 1* and *Original 2*). We also memorize the tuple $v(z_2)$ gathering values of variables indexed by $\mathcal{E}$ when generating Original 2, and $\widetilde{v}(z_1)$ the tuple of values taken by all other endogenous variables on this layer, but when generating Original 1. Assuming the choice of $\mathcal{E}$ identifies a modular structure, $\widetilde{v}(z_1)$ and $v(z_2)$ would encode different aspects of their corresponding generated images, such that one can generate a *hybrid example* mixing these features by assigning the collection of output values of layer $\ell$ with the concatenated tuple $(\widetilde{v}(z_1), v(z_2))$ and feeding it to the downstream part of the generator network.

## 3.3 MEASURING CAUSAL EFFECTS

The above counterfactual hybridization framework allows assessing how a given module $\mathcal{E}$ affects the output of the generator. For this purpose we quantify its causal effect by repetitively generating pairs $(z_1, z_2)$ from the latent space, where both vectors are sampled independently of each other. We then generate and collect hybrid outputs following the above described procedure for a batch of samples and use them to estimate an *influence map* as the mean absolute effect:

$$IM(\mathcal{E}) = \mathbb{E}_{z_2 \sim P(\mathbf{Z})} \mathbb{E}_{z_1 \sim P(\mathbf{Z})} \left| Y_{v(z_2)}^{\mathcal{E}}(z_1) - Y(z_1) \right| \qquad (2)$$

where $Y(z_1) = g_M(z_1)$ is the non-intervened output of the generator for latent input $z_1$. In eq. 2, the difference inside the absolute value can be interpreted as a *unit-level causal effect* in the potential outcome framework (Imbens & Rubin, 2015), and taking the expectation is analogous to computing the *average treatment effect*. Our approach has however two specificities: (1) we take the absolute value of the unit-level causal effects, as their sign may not be consistent across units, (2) the result is averaged over all interventions corresponding to possible values of $z_2$.

While *IM* has the same dimension as the output image, we then average it across color channels to get a single grayscale heat-map pixel map. We also define a scalar quantity to quantify the magnitude of the causal effect, the *individual influence* of module $\mathcal{E}$, by averaging *IM* across output pixels.

## 3.4 UNSUPERVISED DETECTION OF MODULES AND COUNTERFACTUAL IMAGES

A challenge with the hybridization approach is to select the subsets $\mathcal{E}$ to intervene on, especially with networks containing a large amount of units or channels per layer. We use a fine to coarse approach to extract such groups, that we will describe in the context of convolutional layers. First, we estimate *elementary influence maps* (EIM) associated to each individual output channel $c$ of each convolutional layer of the network (i.e. we set $\mathcal{E} = \{c\}$ in eq. (2)). Then influence maps are grouped by similarity to define modules at a coarser scale, as we will describe in detail below.

Representative EIMs for channels of convolutional layers of a VAE trained on the CelebA face dataset (see result section) are shown in Supplementary Fig. 3b and suggest channels are functionally segregated, with for example some influencing finer face feature (eyes, mouth,...) and others affecting the background of the image or the hair. This supports the idea that individual channels can be grouped into modules that are mostly dedicated to one particular aspect of the output.

In order to achieve this grouping in an unsupervised way, we perform clustering of channels using their EIMs as feature vectors as follows. We first pre-process each influence map by: (1) performing a local averaging with a small rectangular sliding window to smooth the maps spatially, (2) thresholding the resulting maps at the 75% percentile of the distribution of values over the image to get a binary image. After flattening image dimensions, we get a (channel×pixels) matrix $\mathbf{S}$ which is then fed to a Non-negative Matrix Factorization (NMF) algorithm with manually selected rank $K$, leading to the factorization $\mathbf{S} = \mathbf{WH}$. From the two resulting factor matrices, we get the $K$ cluster template patterns (by reshaping each rows of $\mathbf{H}$ to image dimensions), and the weights representing the contribution of each of these pattern to individual maps (encoded in $\mathbf{W}$). Each influence map is then ascribed a cluster based on which template pattern contributes to it with maximum weight. The choice of NMF is justified by its success in isolating meaningful parts of images in different components (Lee & Seung, 1999). However, we also compared our approach to the classical k-means clustering algorithm (see Supplementary Fig. 9). In order to further justify our NMF based approach, we also introduce a toy generative model.

**Model 1.** *Consider $\mathbf{Z}$ a vector of $K$ i.i.d. uniformly distributed RVs. Assume a neural network with one hidden layers composed of $m$ vector variables $\mathbf{V}_k$ such that*

$$\mathbf{V}_k = S(\mathbf{H}_k Z_k)\,,$$

*with $\mathbf{H}_k \in \mathbb{R}^n$, $n > 1$ and $S$ a strictly increasing activation function applied entry-wise to the components of each vector (e.g. a leaky ReLU). These endogenous variables are mapped to the output*

$$\mathbf{Y} = \sum_{k=1}^{K} \mathbf{W}^k \mathbf{V}_k\,,$$

*with matrices $\mathbf{W}^k \in \mathbb{R}^{m \times n}$, $m > nK$. Assume additionally the following random choice for the model parameters: (1) all coefficients of $\mathbf{H}_k$'s are sampled i.i.d. from an arbitrary distribution that has a density with respect to the Lebesgue measure, (2) there exists $K$ sets of indices $I_k$ over $[1, m]$ each containing at least one element $l_k \in I_k$ such that for all $j \neq k$, $l_k \notin I_j$, (3) For a given column of $\mathbf{W}^k$, coefficient in $I_k$ are sampled i.i.d. from an arbitrary distribution that has a density with respect to the Lebesgue measure, while the remaining coefficients are set to zero.*

The specific condition on the $I_k$'s enforced in (2) encodes the assumption that there is an area in the image that is only influenced by one of the modules. For example, assuming a simple background/object module pair, it encodes that the borders of the image never belong to the object while the center of the image never belong to background. For this model, we get the following identifiability result.

**Proposition 3.** *For Model 1, with probability 1 we have:*
*(1) The partition of the hidden layer entailed by the $K$ vectors $\{\mathbf{V}_k\}$ corresponds to a disentangled representation (i.e. each vector is modular relatively to the others).*
*(2) Assume influence maps $IM(i, k)$ of each component $i$ in each vector $\mathbf{V}_k$ are known and build the $(m \times nK)$ binary matrix $\mathbf{B}$ by concatenating binary column vectors $\mathbf{B}_{i,k} = IM(i, k) > 0$. Non-negative matrix factorization of $\mathbf{B}$ is unique (up to trivial transformations) and identifies the subsets of endogenous variables associated to each $\mathbf{V}_k$.*

This justifies the use of NMF on a thresholded version of the matrix gathering all EIMs (to generate a binary matrix summarizing their significant influences on each output pixel). Moreover, the application of the sliding window is justified in order to enforce the similarity between the influence maps belonging to the same module, reflected by the condition on identical support $I_k$ for all columns of $\mathbf{W}^k$ in Model 1, and favoring low-rank matrix factorization.

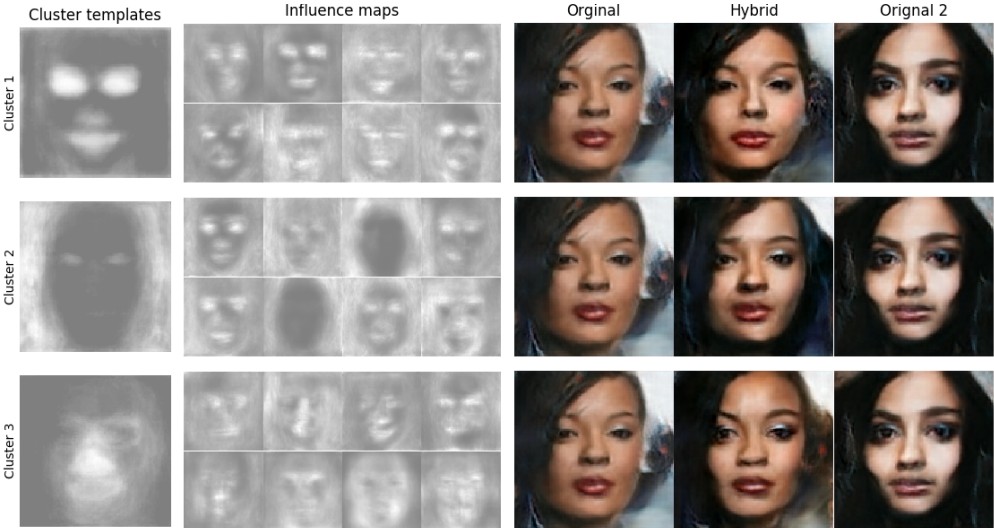

Figure 4: Left: Clustering of influence maps for a BEGAN trained on the CelebA dataset (see text). Center: example EIM of each cluster. Right: samples of the hybridization procedure using as module all channels of the intermediate layer belonging to the cluster of corresponding row. See Fig. 12 for additional samples.

## 4 EXPERIMENTS

### 4.1 DCGAN, $\beta$-VAE AND BEGAN ON THE CELEBA DATASET

We first investigated modularity of genrative models trained on the CelebFaces Attributes Dataset (CelebA)(Liu et al., 2015) and used a basic architecture: a plain $\beta$-VAE [2] (Higgins et al., 2017). We ran the full procedure described in Sec. 3, comprised of EIM calculations, clustering of channels into modules, and hybridization of generator samples using these modules. Hybridization procedures were performed by intervening on the output of the intermediate convolutional layer (indicated in Supplemental Fig. 7). The results are summarized in Supplemental Fig. 8. We observed empirically that setting the number of clusters to 3 leads consistently to highly interpretable cluster templates as illustrated in the figure, with one cluster associated to the background, one to the face and one to the hair. This observation was confirmed by running the following cluster stability analysis: we partition at random the influence maps in 3 subsets, and we use this partition to run the clustering twice on two thirds of the data, both runs overlapping only on one third. The obtained clusters were then matched in order to maximize the label consistency (the proportion of influence maps assigned the same label by both runs) on the overlapping subset, and this maximum consistency was used to assess robustness of the clustering across the number of clusters. The consistency results are provided in Supplemental Fig. 9 and show 3 clusters is a reasonable choice as consistency is large ($> 90\%$) and drops considerably for 4 clusters. Moreover, these results also show that the NMF-based clustering outperforms clustering with the more standard k-means algorithm. In addition, we also assessed the robustness of the clustering by looking at the cosine distance between the templates associated to matching clusters, averaged across clusters. The results, also provided in Supplemental Fig. 9, are consistent with the above analysis with an average cosine similarity of .9 (scalar product between the normalized feature vectors) achieved with 3 clusters (maximum similarity is 1 for perfectly identical templates). Exemplary influence maps shown in Supplemental Fig. 8 (center panel) reflect also our general observation: some maps may spread over image locations reflecting different clusters.

Interestingly, applying the hybridization procedure to the resulting 3 modules obtained by clustering leads to a replacement of the features associated to the module we intervene on, as shown in Supplemental Fig. 8 (center panel), while respecting the overall structure of the image (no discontinuity introduced). For example, on the middle row we see the facial features of the *Original 2* samples are inserted in the *Original 1* image (shown on the left), while preserving the hair.

[2] https://github.com/yzwxx/vae-celebA

While the $\beta$-VAE is designed for extrinsic disentanglement, further work has shown that it can prove suboptimal with respect to other approaches (Chen et al., 2018; Locatello et al., 2018) suggesting further work could investigate whether better extrinsic disentanglement could also favor intrinsic disentanglement. It is however important to investigate intrinsic disentanglement in models for which (extrinsic) disentanglement is not enforced explicitly. This is in particular the case of most GAN-like architectures, who typically outperform VAE-like approaches in terms of sample quality in complex image datasets. Interestingly, the above results could also be reproduced in the official tensorlayer DCGAN implementation[3], equipped with a similar architecture (see Appendix E). This suggests that our approach can be applied to models that have not been optimized for disentanglement. After these experiments with basic models, we used an implementation[4] of the Boundary Equilibrium GAN (BEGAN) (Berthelot et al., 2017), which used to set a milestone in visual quality for higher resolution face images. Most likely due to the increase in the number of layers, we observed that obtaining counterfactuals with noticeable effects required interventions on channels from the same cluster in two successive layers (see Appendix D). The results shown in Fig. 4, obtained by intervening on layers 5 and 6, reveal a clear selective transfer of features from *Original 2* to *Original 1*. As the model was trained on face images cropped with a tighter frame than for the above models, leaving little room for the hair and background, we observe only one module associated to these features (Fig. 4, middle row) showing a clear hair transfer. The remaining two modules are now encoding different aspects of face features: eye contour/mouth/nose for the top row and eyelids/face shape for the bottom row module. We further evaluated the relative quality of the counterfactual images with respect to the original generated images using the Frechet Inception Distance (FID) (Heusel et al., 2017) (Table 2 in the appendix), supporting that the hybridization procedure only mildly affects the image quality, in comparison to the original samples.

## 4.2 BigGAN on the ImageNet dataset

In order to check whether our approach could scale to high resolution generative models, and generalize to complex image datasets containing a variety of objects, we used the BigGAN-deep architecture (Brock et al., 2018), pretrained[5] on the ImageNet dataset[6]. This is a conditional GAN architecture comprising 12 so-called Gblocks, each containing a cascade of 4 convolutional layers (see Appendix C for details). Each Gblock also receives direct input from the latent variables and the class label, and is bypassed by a skip connection. We then checked that we were able to generate hybrids by mixing the features of different classes. As for the case of BEGAN, intervening on two successive layers within a Gblock was more effective to generate counterfactuals (examples are provided for the 7th Gblock). Examples provided in Fig. 5 (cock-ostrich) show that it is possible to generate high quality counterfactuals with modified background while keeping a very similar object in the foreground. In a more challenging situation, with objects of different nature (Koala-teddy bear on the same figure), meaningful combinations of each original samples are still generated: e.g. a teddy bear in a tree (bottom row), or a "teddy-koala" merging teddy texture with the color of a koala on a uniform indoor background with a wooden texture (top row).

In order to investigate how the generated counterfactual images can be used to probe and improve the robustness of classifiers to contextual changes, we compared the ability of several SOTA pretrained classifier available on Tensorflow-hub (`https://tfhub.dev/`, see Appendix C for details) to recognize one of the original classes. Fig. 6 shows the average recognition rate of the most recognized original class (teddy-bear or koala), as a function of layer depth tends overall to increase. We first observe that high recognition rates are in line with the small pixel distance between hybrids and original when intervening at layers closest to the output (right panel). Interestingly, at intermediate blocks 5-6, there is a clear contrast between classifiers, with the Inception resnet performing better than the others. Interestingly, examples of non-consensual classification results in Supplementary Table 3, together with the associated hybrids (Supplementary Fig. 17) suggest different SOTA classifiers rely on different aspects of the image content to take their decision (e.g. background versus object).

---

[3] `https://github.com/tensorlayer/dcgan`

[4] `https://github.com/Heumi/BEGAN-tensorflow`

[5] `https://tfhub.dev/deepmind/biggan-deep-256/1`

[6] `http://www.image-net.org/`

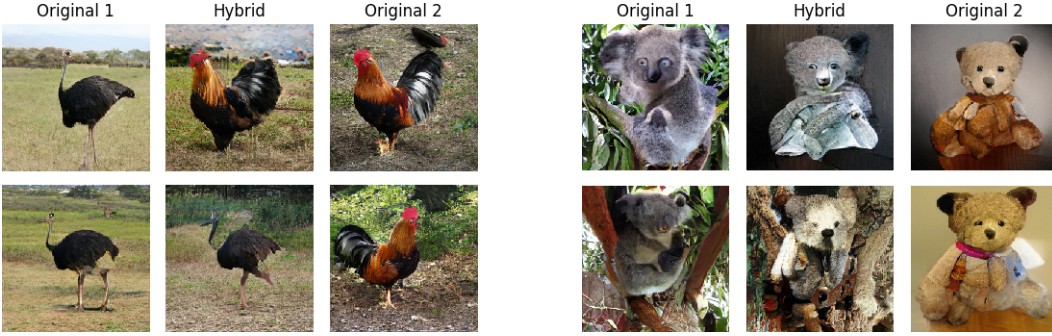

Figure 5: Examples of BigGAN hybridizations across classes. Left: ostrich-cock, right: koala-teddy. See Figs. 13-15 for additional samples and Fig. 14-16 for entropy analysis.

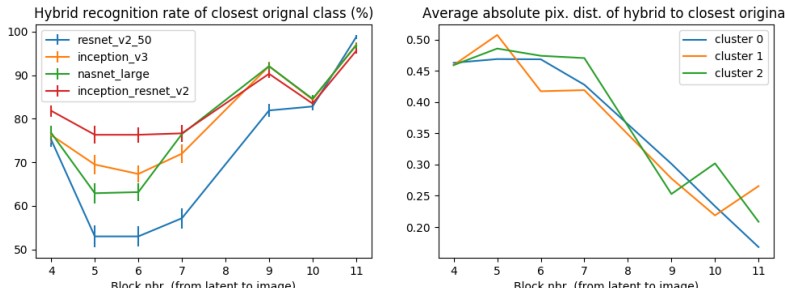

Figure 6: Analysis of classifier robustness to counterfactual changes (bars indicate standard error).

## CONCLUSION

We introduced a mathematical definition of *disentanglement*, related it to the causal notion of counterfactual and used it for the unsupervised characterization of the representation encoded by different groups of channels in deep generative architectures. We found evidence for interpretable modules of internal variables in four different generative models trained on two complex real world datasets. Our framework opens a way to a better understanding of complex generative architectures and applications such as the style transfer (Gatys et al., 2015) of controllable properties of generated images at low computational cost (no further optimization is required), and the automated assessment of robustness of object recognition systems to contextual changes. From a broader perspective, this research direction contributes to a better exploitation of deep neural networks obtained by costly and highly energy-consuming training procedures, by (1) enhancing their interpretability and (2) allowing them to be used for tasks their where not trained for. This offers a perspective on how more sustainable research in Artificial Intelligence could be fostered in the future.

### 4.3 APPENDICES AND IMPLEMENTATIONS

Appendices and supplementary Figures and Tables are available online with the arXiv version of this paper (`https://arxiv.org/abs/1812.03253`). Implementations are available on the companion website `https://gitlab.tuebingen.mpg.de/besserve/deepcounterfactuals`.

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

# Supplementary information

## APPENDIX A: FORMAL DEFINITIONS AND RESULTS FOR CAUSAL GENERATIVE MODELS (SECTION 2)

We rely on the assumption that a trained generator architecture can be exploited as a mechanistic model, such that parts of this model can be manipulated independently. A mathematical representation of such models can be given using *structural causal models (SCMs*, that rely on structural equations (SEs) of the form $Y := f(X_1, X_2, \cdots, X_N, \epsilon)$, denoting the assignment of a value to variable $Y$, computed from the values of other variables $X_k$ in the system under consideration, and of putative exogenous influences $\epsilon$, imposed by factors outside the system under study. As in the above equation, we will use uppercase letters to indicate variables being the outcome of a structural assignment, while specific values taken by them will be lower case. SEs stay valid even if right-hand side variables undergo a change due to interventions (Pearl, 2009; Peters et al., 2017, e.g.), and can model the operations performed in computational graphs of modern neural network implementations. Such graphs then depict SCMs made of interdependent modules, for which assignments' dependencies are represented by a directed acyclic graph $\mathcal{G}$. Without loss of generality, we introduce a Causal Generative Model (CGM) $M$ capturing the computational relations between a selected subset of variables comprising: (1) the input latent variables $\{z_k\}$, (2) the generator's output $Y$ (typically multi-dimensional), and (3) a collection of possibly multi-dimensional endogenous (internal) variables forming an *intermediate representation* such that the generator's output can be decomposed into two successive steps as $\{Z_k\} \mapsto \{V_k\} \mapsto Y$. In a feed-forward neural network, one $V_k$ may for instance represent one channel of the output of a convolutional layer (e.g. after application of the ReLU non-linearity).

**Definition 6** (Causal Generative Model (CGM)). *Given $K$ real-valued* latent *variables $\boldsymbol{z} = (z_k)$ taking arbitrary values on domain $\mathcal{Z} = \prod_{k=1}^{K} \mathcal{Z}_k$, where all $\mathcal{Z}_k$'s are closed intervals, the CGM $M = \mathbb{G}(\mathcal{Z}, \boldsymbol{S}, \mathcal{G})$ comprises a directed acyclic graph $\mathcal{G}$ and a set $\boldsymbol{S}$ of $N+1$ deterministic continuous structural equations that assign:*

- *$N$ endogenous variables $\{V_k := f_k(\boldsymbol{Pa}_k)\}_{k=1..N}$ taking values in Euclidean spaces $(\mathcal{V}^k)_{k=1..N}$, based on their endogenous or latent parents $\boldsymbol{Pa}_k$ in $\mathcal{G}$,[7]*
- *one output $Y := f_y(\boldsymbol{Pa}_y)$ taking values in Euclidean space $\mathcal{Y}$, parents $\boldsymbol{Pa}_y$ in $\mathcal{G}$ being endogenous.*

*Moreover, $z_k$'s are the only sources and $Y$ is the only sink.[8]*

The graph of an example CGM is exemplified on Fig. 2b, consisting of 3 endogenous variables, 2 latent inputs and the output. This aligns with the definition of a deterministic structural causal model by Pearl (2009, chapter 7), once our latent variables are identified with exogenous ones. CGMs have however specificities reflecting the structure of models encountered in practice. For instance, variable assignments may or may not involve latent/exogenous variables in their right-hand side, which is unusual in causal inference. This allows modeling feed-forward networks consisting in a first layer receiving latent inputs followed by a cascade of deterministic operations in downstream layers. The above definition guaranties several basic properties found in the computational graph of existing generative networks: (1) all endogenous variables $V_k$ are unambiguously assigned once $\boldsymbol{z}$ is chosen, (2) the output $Y$ is unambiguously assigned once either $\boldsymbol{z}$ is chosen, or, alternatively, if an appropriate subset of $V_k$'s, such as $\boldsymbol{Pa}_y$, is assigned. This allows us to introduce several useful mappings.

In an ideal case, while the support of the latent distribution covers the whole latent space $\mathcal{Z}$, internal variables and outputs typically live on manifolds of smaller dimension than their ambient space. These can be defined as the images[9] of $\mathcal{Z}$ by operations of the graph: the output image $\mathcal{Y}_M = Y[\mathcal{Z}]$, the endogenous images $\mathcal{V}_M^k = V_k[\mathcal{Z}]$ for a single variable, $\boldsymbol{\mathcal{V}}_M^{\mathcal{E}} = \{\boldsymbol{v} \in \prod_{k \in \mathcal{E}} \mathcal{V}^k : \boldsymbol{v} = (v_k(\boldsymbol{z}))_{k \in \mathcal{E}}, \boldsymbol{z} \in \mathcal{Z}\}$ for a subset of variables indexed by $\mathcal{E}$, and $\boldsymbol{\mathcal{V}}_M$ when $\mathcal{E}$ includes all endogenous variables.

---

[7]$A$ is a parent of (child) $B$ whenever there is an arrow $A \to B$.

[8]Sources are parentless nodes, sinks are childless.

[9]An image $f[A]$ is the subset of outputs of $f$ for a subset A of input values.

Functions assigning $Y$ from latent variables and from endogenous variables, respectively, are

$$g_M : \begin{array}{ccc} \mathcal{Z} & \to & \mathcal{Y}_M, \\ \boldsymbol{z} & \mapsto & Y(\boldsymbol{z}) \end{array} \quad \text{and} \quad \tilde{g}_M : \begin{array}{ccc} \boldsymbol{\mathcal{V}}_M & \to & \mathcal{Y}_M, \\ \boldsymbol{v} & \mapsto & Y(\boldsymbol{v}), \end{array}$$

and we call them *latent* and *endogenous mappings*, respectively. Given the typical choice of dimensions for latent and endogenous variables, the $V_k$'s and $Y$ are constrained to take values in subsets of their euclidean ambient space. We will assume that $\tilde{g}_M$ and $g_M$ define proper embeddings, in particular implying that they are both invertible. We call a CGM satisfying these assumptions an embedded CGM.

With this vocabulary we can for example verify the example of Fig. 2b contains exactly two layers (in green). Note $g_M$ and $\tilde{g}_M$ are well defined because the output can be unambiguously computed from their inputs by successive assignments along $\mathcal{G}$, and are both surjective due to appropriate choices for domains and codomains. All defined image sets ($\boldsymbol{\mathcal{V}}_M^\ell$, $\mathcal{Y}_M$, ...) are constrained by the parameters of $M$, and are typically not easy to characterize. For example $\boldsymbol{\mathcal{V}}_M$ is likely *a strict subset* of the Cartesian product $\prod_k \mathcal{V}_M^k$.

Importantly, the image set $\mathcal{Y}_M$ of a trained model is of particular significance, as it should approximate at best the support of the data distribution we want to model. Learning the generator parameters such that $\mathcal{Y}_M$ precisely matches the support of the target data distribution is arguably a major goal for generative models (see e.g. Sajjadi et al. (2018)).

As we will manipulate properties of the output, we restrict ourselves to transformations that respect the topology of $\mathcal{Y}_M$, and use embeddings as the basic structure for it, allowing inversion of $g_M$.

**Definition 7** (Embedded CGMs). *If $f : X \to Y$ is a continuous injective function with continuous inverse $f^{-1} : f[X] \to Y$, we call $f$ an* embedding *of $X$ in $Y$. We say that a CGM $M$ is* embedded *if $g_M$ and $\tilde{g}_M$ are respective embeddings of $\mathcal{Z}$ and $\boldsymbol{\mathcal{V}}_M$ in $\mathcal{Y}$.*

Since Definition 6 imposes continuous structural equations, which is satisfied for all operations in standard generative models, injectivity of $g_M$ is the key additional requirement for embedded CGMs.

**Proposition 4.** *If $\mathcal{Z}$ of CGM $M$ is compact (all $\mathcal{Z}_k$'s are bounded), then $M$ is embedded if and only if $g_M$ is injective.*

Proof is provided in Appendix B. This implies that generative models based on uniformly distributed latent variables (the case of many GANs), provided they are injective, are embedded CGMs. While VAEs' latent space is typically not compact (due to the use of normally distributed latent variables), we argue that restricting it to a product of compact intervals (covering most of the probability mass) will result in an embedded CGM that approximates the original one for most samples.

Based on this precise framework, we can now provide the formal definitions and results described informally in main text.

The CGM framework allows defining counterfactuals in the network following Pearl (2014).

**Definition 8** (Unit level counterfactual). *Given CGM $M$, for a subset of endogenous variables $\mathcal{E} = \{e_1, .., e_n\}$, and assignment $\boldsymbol{h}$ of these variables, we define the interventional CGM $M_{\boldsymbol{h}}$ obtained by replacing structural assignments for $\mathbf{V}_{|\mathcal{E}}$ by assignments $\{V_{e_k} := h_k(\boldsymbol{z})\}_{e_k \in \mathcal{E}}$. Then for a given value $\boldsymbol{z}$ of the latent variables, called* unit, *the* unit-level counterfactual *is the output of $M_{\boldsymbol{h}}$: $Y_{\boldsymbol{h}}^{\mathcal{E}}(\boldsymbol{z}) = g_{M_{\boldsymbol{h}}}(\boldsymbol{z})$.*

Definition 8 is also in line with the concept of *potential outcome* (Imbens & Rubin, 2015). Importantly, conterfactuals induce a transformation of the output of the generative model.

**Definition 2** (Counterfactual mapping). *Given an embedded CGM, we call the continuous map*

$$\overset{\frown}{Y}_{\boldsymbol{h}}^{\mathcal{E}} : y \mapsto Y_{\boldsymbol{h}}^{\mathcal{E}}\left(g_M^{-1}(y)\right)$$

*the $(\mathcal{E}, \boldsymbol{h})$-counterfactual mapping. We say it is* faithful *to $M$ whenever $\overset{\frown}{Y}_{\boldsymbol{h}}^{\mathcal{E}}[\mathcal{Y}_M] \subset \mathcal{Y}_M$.*

Our approach then relates counterfactuals to a form of disentanglement allowing transformations of the internal variables of the network as follows.

**Definition 3** (Intrinsic disentanglement). *In a CGM $M$, endomorphism $T : \mathcal{Y}_M \to \mathcal{Y}_M$ is intrinsically disentangled with respect to a subset $\mathcal{E}$ of endogenous variables, if it exists a transformation $T'$ of endogenous variables such that for any latent $z \in \mathcal{Z}$, leading to the tuple of values $v \in \mathcal{V}_M$,*

$$T(Y(v)) = Y(T'(v)) \tag{3}$$

*where $T'(v)$ only affects the variables indexed by $\mathcal{E}$.*

In this definition, $Y(v)$ corresponds to the unambiguous assignment of $Y$ based on endogenous values.[10] Fig. 2d illustrates this second notion of disentanglement, where the split node indicates that the value of $V_2$ is computed as in the original CGM (Fig. 2b) before applying transformation $T^2$ to the outcome.

Intrinsic disentanglement relates to a causal interpretation of the generative model's structure in the sense that it expresses a form of robustness to perturbation of one of its subsystems. Counterfactuals represent examples of such perturbations, and as such, may be disentangled given their faithfulness.

**Proposition 1** (Counterfactuals and disentanglement, formal). *For an embedded CGM $M$ and continuous assignment $h$, the $(\mathcal{E}, h)$-counterfactual mapping $\overset{\curvearrowright \mathcal{E}}{Y}_h$ is faithful if and only if it is intrinsically disentangled with respect to subset $\mathcal{E}$. Moreover, if $h[\mathcal{Z}] \subset \mathcal{V}_M^{\mathcal{E}}$, it is sufficient that $\mathcal{E}$ and $\bar{\mathcal{E}}$ do not have common latent ancestors for $\overset{\curvearrowright \mathcal{E}}{Y}_h$ to be faithful.*

The proof is provided in Appendix B.

## Appendix B: Additional details for Section 2 and 3

### Topological concepts

**Continuity.** We recall the classical definition of continuity of $f : X \to Y$. Given the respective topologies $\tau_X$ and $\tau_Y$ of domain and codomain (the sets of all open sets), $f$ is continuous whenever for all $A \in \tau_Y$, $f^{-1}(A) \in \tau_X$.

**Euclidean topology.** For defining continuity between Euclidean spaces, we rely on the Euclidean (or standard) topology naturally induced by the metric: as set is open if and only if it contains and open ball around each if its points.

**Subset topology.** When restricting the domain or codomain of a mapping to a subset A of the Euclidean space, we rely on the subspace topology, that consists in the intersection of A will all open sets.

### Proof of Proposition 4

Following a result stated in Armstrong (2013), since $\mathcal{Z}$ is compact and the codomain of $g_M$ is Hausdorff (because Euclidean), then a continuous (by definition) and injective $g_M$ is an embedding. In addition, $g_M$ injective implies $\tilde{g}_M^\ell$'s are injective on their respective domains $\mathcal{V}_M^\ell$. Moreover, the $\mathcal{V}_M^\ell$'s being image of a compact $\mathcal{Z}$ by a continuous mapping (by the CGM definition), they are compact, such that the respective $\tilde{g}_M^\ell$'s are also embeddings.

### Proof of Proposition 1

**Part 1: Proof of the equivalence between faithful and disentangled.** One conditional is trivial: if a transformation is disentangled, it is by definition an endomorphism of $\mathcal{Y}_M$ so the counterfactual mapping must be faithful.

For the second conditional, let us assume a faithful $\overset{\curvearrowright \mathcal{E}}{Y}_h$ and denote the (unambiguous) map from $\mathcal{V}$ to the output

$$Y_M : \begin{array}{ccc} \mathcal{V}^\ell & \to & \mathcal{Y} \\ v & \mapsto & Y(v) \end{array} \ .$$

---

[10]Note the mapping $v \mapsto Y(v)$ differs from $\tilde{g}_M^\ell$ because its domain is not restricted to the image set $\mathcal{V}_M^\ell$.

This map differs from $\tilde{g}_M$ due to its broader domain and codomain, such that it is neither necessarily an injection nor a surjection, but they coincide on the image $\mathcal{V}_M$. We can first notice (using Definition 8 and the embedding property) that the counterfactual mapping can be decomposed as

$$\overset{\frown}{Y}{}^{\mathcal{E}}_{\boldsymbol{h}} = Y_M \circ T' \circ (\tilde{g}_M)^{-1}\,, \quad \text{where} \quad T' : \begin{array}{ccc} \mathcal{V}^{\bar{\mathcal{E}}} \times \mathcal{V}^{\mathcal{E}} & \to & \mathcal{V}^{\bar{\mathcal{E}}} \times \mathcal{V}^{\mathcal{E}} \\ (\tilde{\boldsymbol{v}},\, \boldsymbol{v}) & \mapsto & (\tilde{\boldsymbol{v}},\, \boldsymbol{h}) \end{array}\,.$$

Since $\overset{\frown}{Y}{}^{\mathcal{E}}_{\boldsymbol{h}}$ is faithful, it is then an endomorphism of it $\mathcal{Y}_M$ (continuity comes form the composition of continuous functions), as required by the definition of disentanglement.

For any $\boldsymbol{v} \in \mathcal{V}$, consider then the quantity

$$\overset{\frown}{Y}{}^{\mathcal{E}}_{\boldsymbol{h}}(Y(\boldsymbol{v})) = \overset{\frown}{Y}{}^{\mathcal{E}}_{\boldsymbol{h}} \circ \tilde{g}_M(\boldsymbol{v})\,,$$

using the above decomposition, we can rewrite it as

$$\overset{\frown}{Y}{}^{\mathcal{E}}_{\boldsymbol{h}}(Y(\boldsymbol{v})) = Y_M \circ T' \circ (\tilde{g}_M)^{-1} \circ \tilde{g}_M(\boldsymbol{v}) = Y_M \circ T'(\boldsymbol{v})\,,$$

where $T'$ is a transformation that only affects endogenous variables in $\mathcal{E}$, demonstrating that $\overset{\frown}{Y}{}^{\mathcal{E}}_{\boldsymbol{h}}(Y(\boldsymbol{v}))$ is disentangled with respect to $\mathcal{E}$.

**Part 2: Sufficient condition.** This is a direct application of the following Proposition 5 after observing that in our case, the endomorphism $T^{\mathbb{E}}$ required in this proposition is the constant function with value $\boldsymbol{h}$.

**Proposition 5.** *For embedded CGM $M$, if a subset $\mathcal{E}$ of endogenous variables does not share common latent ancestors[11] with the reminder of endogenous variables $\bar{\mathcal{E}}$, then any endomorphism $T^{\mathcal{E}} : \mathcal{V}^{\mathcal{E}}_M \to \mathcal{V}^{\mathcal{E}}_M$ leads to a transformation*

$$T : y \mapsto \tilde{g}_M \circ T' \circ (\tilde{g}_M)^{-1}(y)\,,$$

$$\text{with} \quad T' : \begin{array}{ccc} \mathcal{V}^{\bar{\mathcal{E}}}_M \times \mathcal{V}^{\mathcal{E}}_M & \to & \mathcal{V}^{\bar{\mathcal{E}}}_M \times \mathcal{V}^{\mathcal{E}}_M\,, \\ (\tilde{\boldsymbol{v}},\, \boldsymbol{v}) & \mapsto & (\tilde{\boldsymbol{v}},\, T^{\mathcal{E}}(\boldsymbol{v}))\,, \end{array}$$

*such that $T$ is disentangled with respect to $\mathcal{E}$ in $M$.*

*Proof of Proposition 5.* The absence of common latent ancestor between $\bar{\mathcal{E}}$ and $\mathcal{E}$ ensures that values in both subsets are unambiguously assigned by non-overlapping subsets of latent variables, $A$ and $B$ respectively, such that we can write

$$\left(\boldsymbol{V}_{|\bar{\mathcal{E}}},\, \boldsymbol{V}_{|\mathcal{E}}\right) = (f_A(\boldsymbol{z}_A),\, f_B(\boldsymbol{z}_B))$$

This implies that the image set of this layer fully covers the Cartesian product of the image sets of the two subsets of variables, i.e. $\mathcal{V}_M = \mathcal{V}^{\bar{\mathcal{E}}}_M \times \mathcal{V}^{\mathcal{E}}_M$, and guaranties that $T'$ is and endomorphism of $\mathcal{V}^{\ell}_M$ for any choice of endomorphism $T^{\mathcal{E}}$. This further implies $T$ is well defined and an endomorphism. $\square$

PROOF SKETCH FOR PROPOSITION 3

Due to the i.i.d. assumption for components of $\boldsymbol{Z}$ and the structure following the sufficient condition of Prop. 1, it is clear that the subsets of endogenous variables associated to each $\boldsymbol{V}_k$ are modular and the associated partition of the hidden layer is a disentangled representation. The choice of increasing dimensions as well as the i.i.d. sampling of the model parameters from a distribution with a density make sure the resulting mapping is injective (and hence follow the embedded CGM assumptions of Def. 7) and that counterfactual hybridization of any component of $\boldsymbol{V}_k$ will result in an influence map whose support covers exactly $I_k$. Finally, the conditions on the $I_k$'s and the thresholding approach guaranties a rank $K$ binary factorization of the matrix $B$, with one factor gathering the indicator vectors associated to each $\boldsymbol{V}_k$ and the uniqueness of this factorization is guaranteed by classical NMF identifiability results, e.g. following (Diop et al., 2017)[Theorem III,1].

[11] $A$ is an ancestor of $B$ whenever there is a directed path $A \to .. \to B$ in the graph

## APPENDIX C: ARCHITECTURE DETAILS

### VANILLA $\beta$-VAE AND DCGAN

The $\beta$-VAE architecture is presented in Supplementary Fig. 7 and is very similar to the DCGAN architecture. Hyperparameters for both structures are specified in Table 1.

### BEGAN CELEBA

We used the method proposed in Berthelot et al. (2017) for CelebA dataset. We used the pre-trained model with the same architecture as was used in the paper. It consists of three blocks of convolutional layers each followed by an upsampling layer. The filter size of convolutional layers is $(3, 3)$ all over the generator. There is also skip connections in the model that is argued to increase the sharpness of images. Consult (Berthelot et al., 2017, Figure 1) for architectural details.

### BIGGAN-DEEP-256 ARCHITECTURE DETAILS

The pretrained model is taken from *Tensorflow-hub* (https://tfhub.dev/, we summarize below the main aspects of the architectures. We used the BigGan-deep architecture of Brock et al. (2018) as a pre-trained model on 256x256 ImageNet. We did not retrain the model. The architecture consists of several ResBlocks which are the building block of the generator. Each ResBlock contains BatchNorm-ReLU-Conv Layers followed by upsampling transformations and augmented with skip connections that bring fresh signal from the input to every ResBlock. Consult Brock et al. (2018) for architectural details.

### CLASSIFIERS ARCHITECTURE DETAILS

All pretrained models are taken from *Tensorflow-hub* (https://tfhub.dev/, we summarize below the main aspects of the architectures.

### INCEPTION_RESNET_V2

Inception ResNet V2 is a neural network architecture for image classification, originally published by

> Christian Szegedy, Sergey Ioffe, Vincent Vanhoucke, Alex Alemi: "Inception-v4, Inception-ResNet and the Impact of Residual Connections on Learning", 2016.

### INCEPTION_V3

Inception V3 is a neural network architecture for image classification, originally published by

> Christian Szegedy, Vincent Vanhoucke, Sergey Ioffe, Jonathon Shlens, Zbigniew Wojna: "Rethinking the Inception Architecture for Computer Vision", 2015.

### NASNET_LARGE

NASNet-A is a family of convolutional neural networks for image classification. The architecture of its convolutional cells (or layers) has been found by Neural Architecture Search (NAS). NAS and NASNet were originally published by

> Barret Zoph, Quoc V. Le: "Neural Architecture Search with Reinforcement Learning", 2017. Barret Zoph, Vijay Vasudevan, Jonathon Shlens, Quoc V. Le: "Learning Transferable Architectures for Scalable Image Recognition", 2017.

NASNets come in various sizes. This TF-Hub module uses the TF-Slim implementation nasnet_large of NASNet-A for ImageNet that uses 18 Normal Cells, starting with 168 convolutional filters (after the "ImageNet stem"). It has an input size of 331x331 pixels.

RESNET_V2_50

ResNet V2 is a family of network architectures for image classification with a variable number of layers. It builds on the ResNet architecture originally published by

> Kaiming He, Xiangyu Zhang, Shaoqing Ren, Jian Sun: "Deep Residual Learning for Image Recognition", 2015.

The full preactivation 'V2' variant of ResNet used in this module was introduced by

> Kaiming He, Xiangyu Zhang, Shaoqing Ren, Jian Sun: "Identity Mappings in Deep Residual Networks", 2016.

## APPENDIX D: IMPLEMENTATION DETAILS.

The file influence.py provided at the link `https://www.dropbox.com/sh/4qnjictmh4a2soq/AAAa5brzPDlt69QOc9n2K4uOa?dl=0` contains key elements of the code for counterfactual analysis of generative models.

## APPENDIX E: ADDITIONAL RESULTS

### INFLUENCE MAP CLUSTERING AND HYBRIDIZATION IN DCGANS

We replicated the above approach for GANs on the CelebA dataset. The result shown in Supplemental Fig. 11 summarize the main differences. First, the use of three clusters seemed again optimal according to the stability of the obtained cluster templates. However, we observed that the eyes and mouth location were associated with the top of the head in one cluster, while the rest of the face and the sides of the image (including hair and background) respectively form the two remaining clusters. In this sense, the GAN clusters are less aligned with high level concepts reflecting the causal structure of these images. However, such clustering still allows a good visual quality of hybrid samples.

## APPENDIX F: ADDITIONAL DISCUSSION

### EXTENSION AND RELATION TO OTHER FRAMEWORKS

While we defined disentanglement of transformations, this concept has been classically attributed to a representation, or factors of variation. Focusing on a transformation aligns to our aim of providing an agnostic definition, in the sense that it only relies on the generator architecture and a given transformation, but neither on data, nor on properties that are not stated explicitly in the definition. In contrast, disentangled representation is usually understood as intervening on meaningful/interpretable features in the image, which may be subjective, or at least referring to some external knowledge. Alternatively, one might expect that what should be disentangled is a property or a factor of variation (say "hair color"). We argue that we can state a property is disentangled by the generalizing our notion of disentanglement to a family of transformations as follows.

**Definition 9** (Disentangled family). *A family of transformations (possibly parametric) is disentangled if all members are disentangled with respect to the same $\mathcal{E}$.*

Then a property may be disentangled if the class of all transformations changing "only" the value of this specific property is disentangled according to Definition 9. Finally, one might also expect that several transformations (or families of transformations) should be disentangled with respect to each other, e.g. allowing to state that hair color should be disentangled from hair length. This *relative* disentanglement is easily defined based on our original definition.

**Definition 10** (Relative disentanglement). *The families of functions $(\mathcal{F}_i)_{i \in 1..N}$ are called jointly disentangled whenever they are disentangled with respect to non-overlapping subsets of variables $(\mathcal{E}_i)_{i \in 1..N}$ within a given layer.*[12]

---

[12]or within the latent space for extrinsic disentanglement

We argue that the notion introduced in Higgins et al. (2018) can be framed as a special case of Definition 10.

## SUPPLEMENTARY FIGURES AND TABLES

| Architecture | VAE CelebA | GAN CelebA |
|---|---|---|
| Nb. of deconv. layers/channels of generator | 4/(64,64,32,16,3) | 4/(128,64,32,16,3) |
| Size of activation maps of generator | (8,16,32,64) | (4,8,16,32) |
| Latent space | 128 | 150 |
| Optimization algorithm | Adam ($\beta = 0.5$) | Adam ($\beta = 0.5$) |
| Minimized objective | VAE loss (Gaussian posteriors) | GAN loss |
| batch size | 64 | 64 |
| Beta parameter | 0.0005 | NA |

Table 1

Table 2: FID analysis of BEGAN hybrids. Distance between different pairs of classes (R: Real data, G: Generated data, Ck: Hybrids by intervention on cluster k). The distances are is computed for balanced number of examples ($10k$) for each class and normalized by the FID between the real data and the generated data. It can be seen in the table that Hybrids have a small distance to the generated and also to each other. This can be interpreted as closeness of the distribution of Hybrids to that of generated data suggesting that Hybridization produces visually plausible images.

| | | | | | |
|---|---|---|---|---|---|
| G | 1 | 0 | | | |
| C0 | 1.022 | 0.036 | 0 | | |
| C1 | 1.143 | 0.093 | 0.096 | 0 | |
| C2 | 1.158 | 0.097 | 0.099 | 0.045 | 0 |
| | R | G | C0 | C1 | C2 |

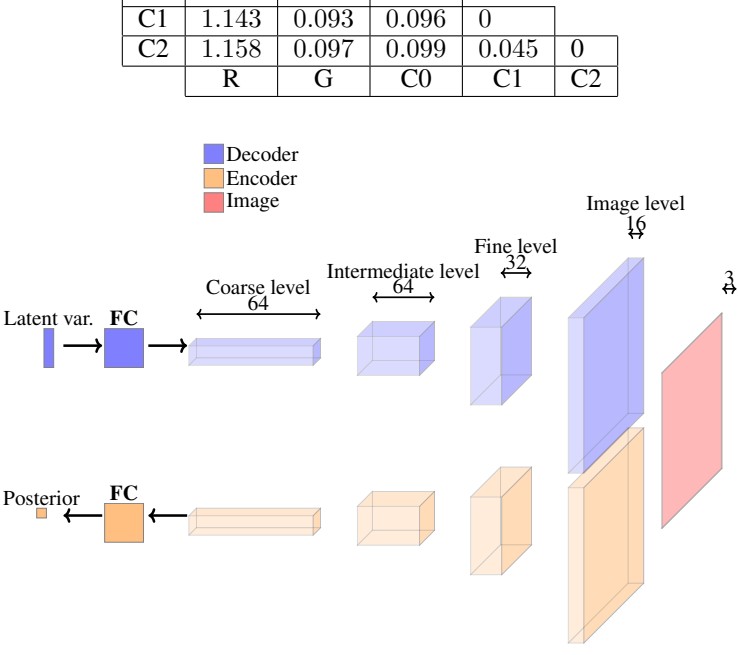

Figure 7: **VAE architecture.** FC indicates a fully connected layer, $z$ is a 100-dimensional isotropic Gaussian vector, horizontal dimensions indicate the number of channels of each layer. The output image size is $64 \times 64$ (or $32 \times 32$ for cifar10) pixels and these dimensions drop by a factor 2 from layer to layer. (reproduced from (Radford et al., 2015).

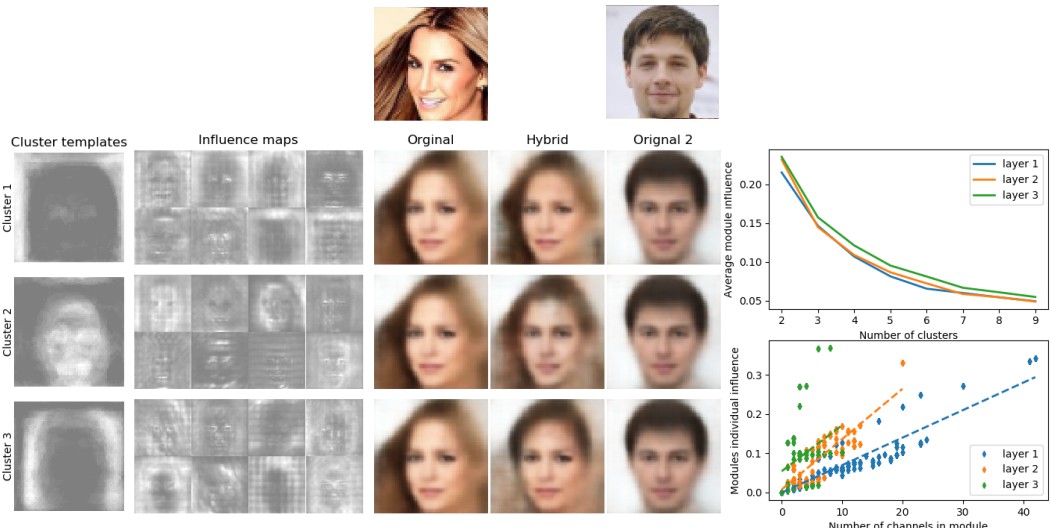

Figure 8: Left: Clustering of influence maps for a VAE trained on the CelebA dataset (see text).. Center: samples of the hybridization procedure using as module all channels of the intermediate layer belonging to the cluster of corresponding row, top insets indicate the original ImageNet pictures use to produce both samples by feeding them to the VAE's encoder. Right: magnitude of causal effects. (top: average influence of modules derived from clustering as a function of the number of clusters; bottom: individual influence of each modules, as a function of the number of channels they contain, dashed line indicate linear regression).

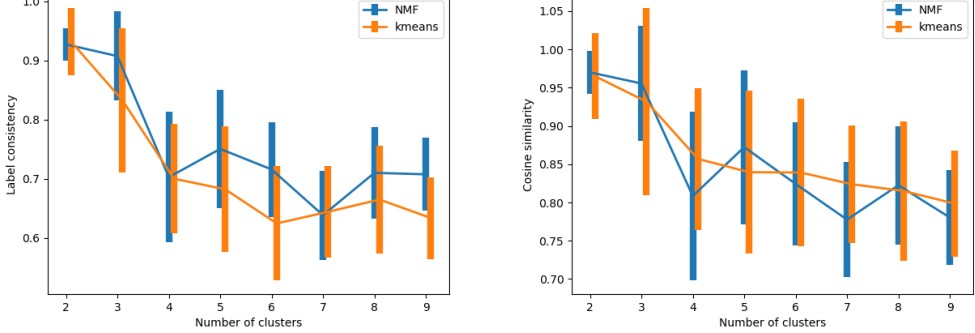

Figure 9: Label consistency (left) and cosine similarity (right) of the clustering of influence maps for the NMF and k-means algorithm. Errorbars indicate standard deviation across 20 repetitions.

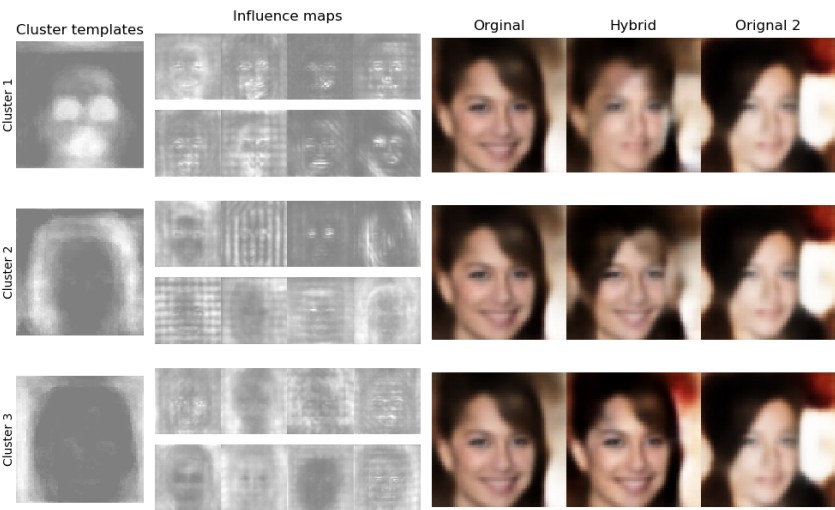

Figure 10: Clustering of influence maps and generation of hybrid samples for a VAE trained on the CelebA dataset (see text).

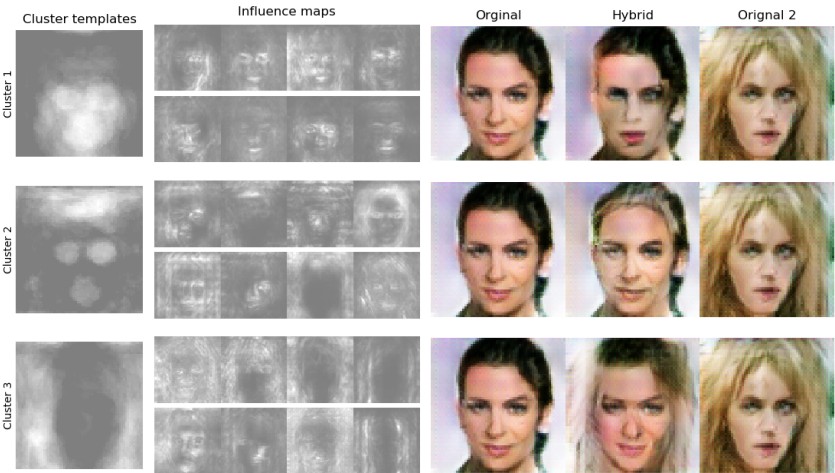

Figure 11: Clustering of influence maps and generation of hybrid samples for a GAN trained on the CelebA dataset (see text).

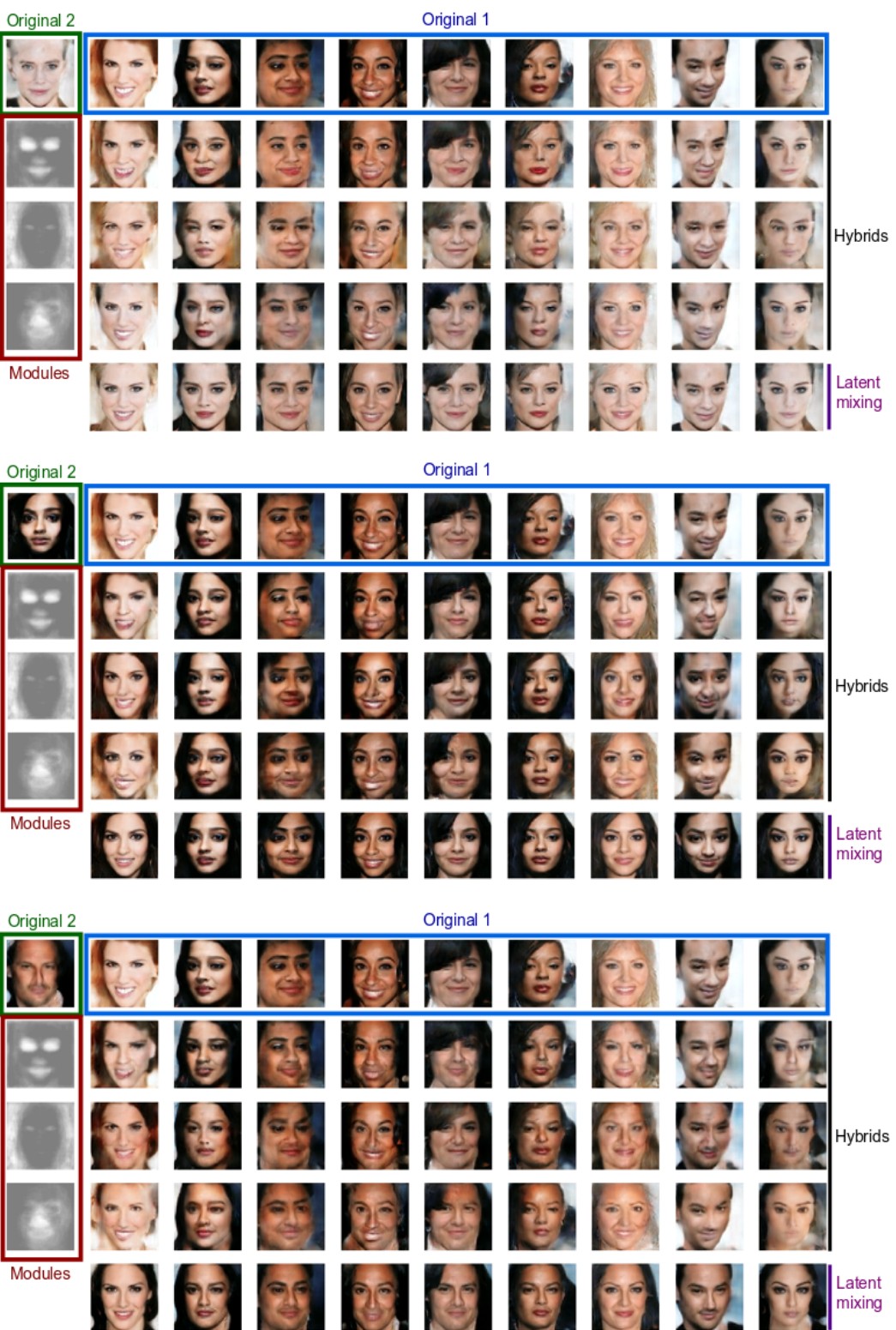

Figure 12: Larger collection of hybrids for the BEGAN. Modules are fixed and extracted by the NMF algorithm using 3 clusters. For each three panel, the "Original 2" sample used to perform the intervention is fixed (show in the top-left corner) while Original 1 varies along the columns.

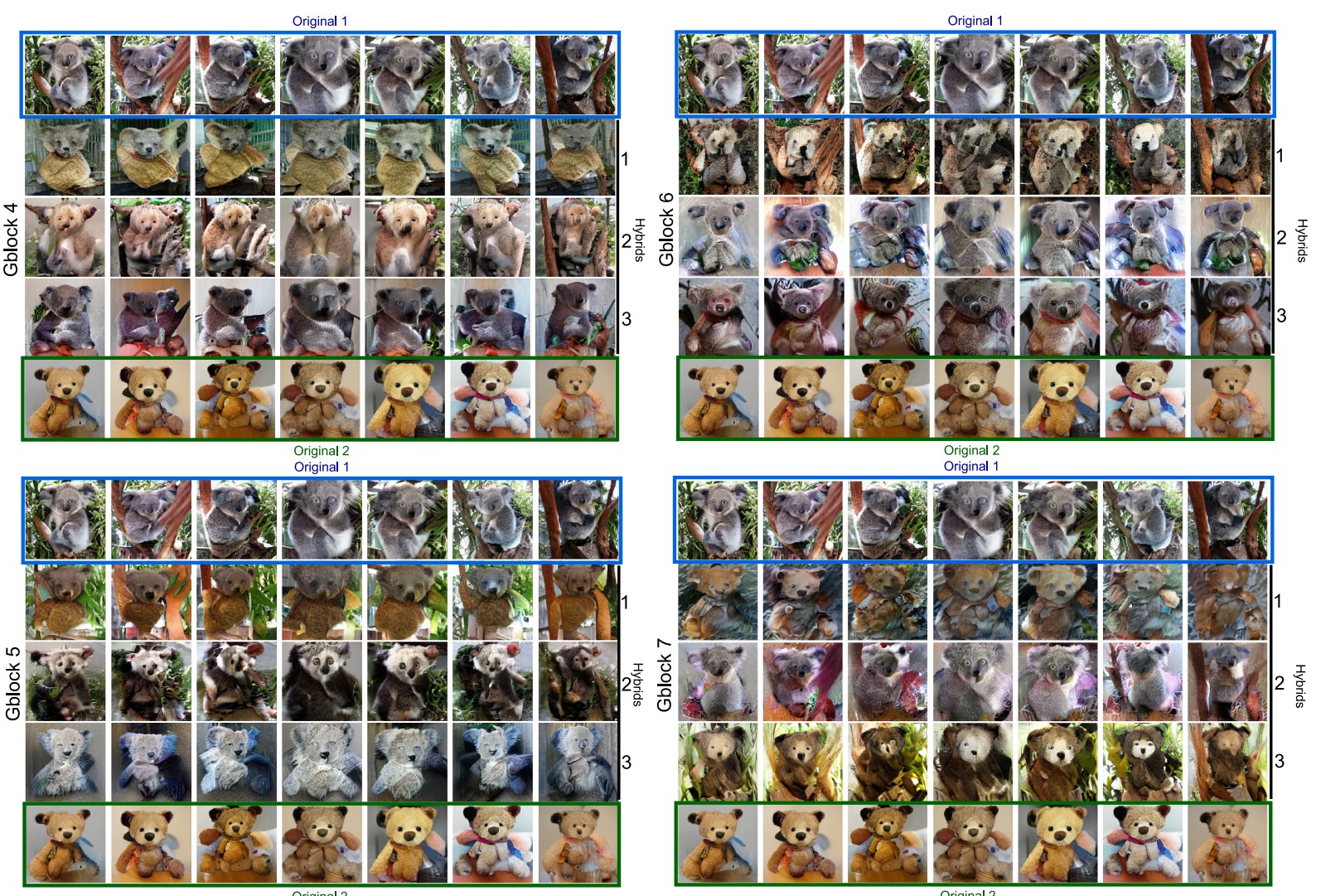

Figure 13: Larger collection of hybrids for the BIGAN, between classes "teddy" and "koala". Each panel corresponds to intervening on a different Gblock (from 4 to 7). Modules are fixed and extracted by the NMF algorithm using 3 clusters. Each row of hybrids corresponds to interventions on one of the extracted module, numbered on the right-hand side.

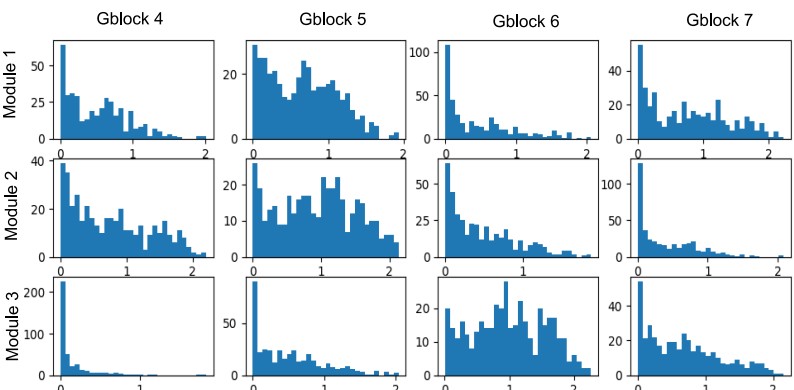

Figure 14: Histogram of the entropy of the resnet_v2_50 classifier logits corresponding to the experiment of Fig. 13. Columns indicate intervened Gblock (from 4 to 7), rows indicate module (as ordered in Fig. 13). The entropy is computed using the probabilistic output for the 10 classes receiving top ranking across all hybrids, normalized to provide a total probability of 1. In particular for Gblock 6 (left column) we can see that the module with poorer quality leads to larger entropy values. Interestingly, entropy values are also much smaller for hybrids based on interventions on the (more abstract level) Gblock number 4. Overall, the results suggests that object texture, which is well rendered in hybrids generated from Gblock 4, is a key information for the classifier's decision.

Figure 15: Larger collection of hybrids for the BIGAN, between classes "cock" and "ostrich". Each panel corresponds to intervening on a different Gblock (from 4 to 7). Modules are fixed and extracted by the NMF algorithm using 3 clusters. Each row of hybrids corresponds to interventions on one of the extracted module, numbered on the right-hand side.

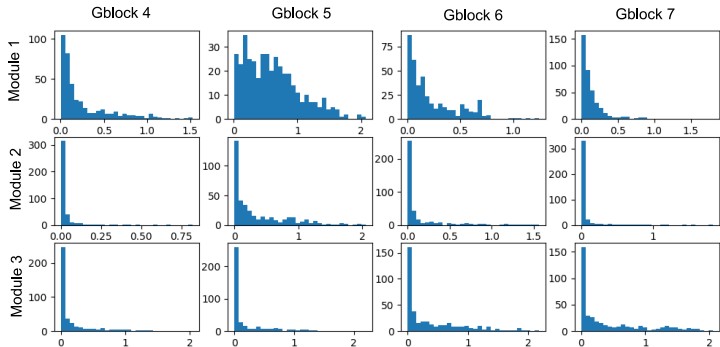

Figure 16: Histogram of the entropy of the resnet_v2_50 classifier logits corresponding to the experiment of Fig. 15. Columns indicate intervened Gblock (from 4 to 7), rows indicate module (as ordered in Fig. 15). The entropy is computed using the probabilistic output for the 10 classes receiving top ranking across all hybrids, normalized to provide a total probability of 1. Interestingly, large entropy is obtained for first module of Gblock 5 (middle column), consistent with the fact the intervention generates a hybrid bird, mixing shape properties of both cock and ostrich.

| | | resnet_v2_50 | inception_v3 | nasnet_large | inception_resnet_v2 |
|---|---|---|---|---|---|
| Left Image | Model | resnet_v2_50 | inception_v3 | nasnet_large | inception_resnet_v2 |
| | Output | koala | koala | koala | koala |
| Middle Image | Model | resnet_v2_50 | inception_v3 | nasnet_large | inception_resnet_v2 |
| | Output | koala | koala | teddy | teddy |
| Right Image | Model | resnet_v2_50 | inception_v3 | nasnet_large | inception_resnet_v2 |
| | Output | koala | teddy | teddy | koala |

Table 3: The classification outcome of several discriminative models for three randomly chosen koala+teddy hybrids (see Figure. 17). The purpose of this experiment is to investigate the use of the proposed intervention procedure for assessing robustness of classifiers. As can be seen in the following images, the resultant hybrids are roughly a teddy bear in a koala context. An ideal classifier must be sensitive to the object present in the scene not the contextual information. A teddy bear must still be classified as a teddy bear even if it appears on a tree which is the koala environment in most of the koala images in the ImageNet dataset. It can be seen in the following table that nasnet_large is more robust to the change of context compared to other classifiers.

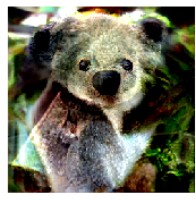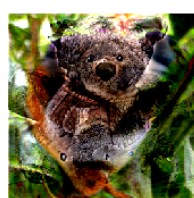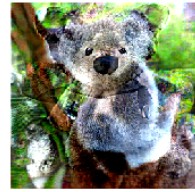

Figure 17: Three koala+teddy hybrids as the inputs to the classifiers of Table. 3

