# OpenReview forum: "Counterfactuals uncover the modular structure of deep generative models"
_ICLR.cc/2020/Conference — Accept (Poster)_

### Official Review · AnonReviewer3 · 2019-10-23
**Official Blind Review #3**

**Rating:** 8

**Review:**

The paper presents a means to uncover the modular structure of deep generative models of images using counterfactuals and presenting evidence for the fact that there are interpretable modules within current popular generative models. The paper is extremely well written with a good balance between mathematical notation and intuitive explanations. I think this paper should certainly be accepted as it provides an interesting and rigorous tool to understand the behavior and properties of deep generative.

I have a few questions and comments:

1) How early does this sort of modularity arise over the course of training? Does it vary for GANs versus beta-vae like models?

2) I think it would be good to contrast this approach with ones that also have latent mixing strategies such as [1, 2] - these do not uncover latent modular structure, but can also produce hybridized images via linear latent space mixing unlike counterfactual assignment of latent vectors like in your work.

3) From what I gather, this approach cannot produce hybridizations between two explicitly provided samples from a dataset, since the models you consider do not have an inference network like in BiGAN or ALI to get the corresponding z vector for a specific image. If this were available, it would be interesting to study influence maps estimated by taking the expectation over pairs of images rather than z-space vectors.

4) Can we quantify modularity or the extent of disentanglement of internal representations under such a framework?

References

[1] Manifold Mixup - https://arxiv.org/pdf/1806.05236.pdf
[2] Adversarial Mixup Resynthesis - https://arxiv.org/pdf/1903.02709v3.pdf

Minor:

Proposition 2 - typo - “then and transformation applied to it”

**Experience Assessment:**

I have read many papers in this area.

**Review Assessment: Checking Correctness Of Derivations And Theory:**

I assessed the sensibility of the derivations and theory.

**Review Assessment: Checking Correctness Of Experiments:**

I assessed the sensibility of the experiments.

**Review Assessment: Thoroughness In Paper Reading:**

I read the paper at least twice and used my best judgement in assessing the paper.

---

> ### Author Response · Authors · 2019-11-15
> **Reply to Rev#3**
>
> Thank you for your useful comments and positive review of our work. We uploaded a revision of the manuscript based on all reviewers’ comments and address yours specifically in this reply.
>
> 1) “How early does this sort of modularity arise over the course of training? Does it vary for GANs versus beta-vae like models? “
> We agree this is an interesting question but a hard one to address quantitatively, given that the sample quality is a critical factor for disentanglement that also evolves during training. We will provide an additional figure on this for the final version.
>
> 2) “I think it would be good to contrast this approach with ones that also have latent mixing strategies such as [1, 2] - these do not uncover latent modular structure, but can also produce hybridized images via linear latent space mixing unlike counterfactual assignment of latent vectors like in your work. “
> In the novel Figure 12 (in appendix), we now provide a large number of hybrids for the BEGAN trained on CelebA, as well as the latent mixing corresponding to averaging the latent vector of the two original samples. It can be observed the intervening on each module provides a different transformation, targeting different aspects of the face (head shape, eyes/mouth, nose/face shape), while mixing in the latent space (according to subjective visual evaluation) tends to find intermediate values for all properties of the image.
>
> 3) “From what I gather, this approach cannot produce hybridizations between two explicitly provided samples from a dataset […] If this were available, it would be interesting to study influence maps estimated by taking the expectation over pairs of images rather than z-space vectors."
> We did sample from original images to illustrate the behavior of the beta-VAE (we updated Fig. 8 in appendix by showing the two CelebA images corresponding to the two samples). As long as an encoder network is available, it can be exploited in this way. We agree this can be interesting if one would want to exploit additional label information in a dataset to look at specific changes in the data. We did not do it in the context of this paper, focused on unsupervised identification of modules, and also because the models that gave the best sample quality were GAN-like and thus deprived from encoder.
>
>  4) “Can we quantify modularity or the extent of disentanglement of internal representations under such a framework? “
> In principle we could elaborate an “absolute” measure of disentanglement by quantifying how far away from the original image set the datapoints are brought. Implementing such absolute measure has been left to future work, as it requires some way to quantify the distance to the manifold generated by the network. Computing such distance is computationally expensive and typically requires numerical approximation (using techniques such as gradient descent to find the closest point on the manifold). Our analyses elaborate several proxies for the modularity quantification, either computed using FID with respect to training data (Table  2), or the consistency of the decision of a classifier trained on the same data (Fig. 5). These disentanglement measures are thus “relative”, as they rely on another representation (provided by a discriminative network) as a reference to quantify how valid are the counterfactual outputs.
>
> In order to increase quantitative assessment of the disentanglement, we provide on Figs.14 and 16 an additional evaluation of the hybrid/disentanglement quality based on the entropy of the probabilitic output of a state of the art classifier. This idea is that good hybrids should have naturalistic feature that allow the classifier to descide of the type of object unambiguously. This measure (as shown when comparing the histagrams to hybrid samples provided in Figs. 13 and 15) can be used to compare the disentanglement of transformations based on different modules and included in different layers (in this case Gblocks) of the architecture.

---

### Official Review · AnonReviewer4 · 2019-11-01
**Official Blind Review #4**

**Rating:** 3

**Review:**

The ideas presented in the paper are interesting and original. Whereas the theory presented has a lot of potential, it seems that the clarity of the paper could be greatly improved, in particular I would have liked more of the formal theory to be included in the body of the paper instead of relying only on the appendix. This especially matters since the theoretical aspect is a key contribution of the paper and the experimental section remains on the light side (it presents mostly examples and lacks more extensive results).

I find the introduction of the proposed definition of disentanglement in sections 2.2/2.3 confusing. The authors first define “extrinsic” disentanglement of a transformation in the data space as corresponding to a transformation of one dimension only in the latent space. In section 2.3 a transformation is called “intrinsically” disentangled if it corresponds to a transformation of a subset of variables in the space of endogenous variables. It should be made clearer from the start that disentanglement is here only a property of the transformations and that the authors are not trying to define a disentangled internal representation. Further, some important questions like how to choose the reference endogenous variables and how to choose the subset E are left entirely to the experiments section. The definition of disentanglement proposed is however tied to these choices and a quick discussion would be helpful.

Whereas the theory from section 2 seems precise and formal (at least in the appendix, although I did not check all the proofs), the procedure to identify modules comes with no guarantees and relies on several choices: local averaging, thresholding, nbr of clusters (the choice of this one is in my opinion well justified). Taking that into account a more extensive experimental validation would be needed to demonstrate that modules can be reliably identified. The results presented on CelebA and ImageNet are interesting, in particular using different models is a good idea, however the evaluation relies mostly on a few cases or examples and I would have liked to see more quantitative results, e.g. like in Figure 8 Appendix F.

Note on related work:
It has been shown (Isolating Sources of Disentanglement in VAEs by Duvenaud et al., Disentangling by Factorising by Kim et al., Challenging Common Assumptions in the Unsupervised Learning of Disentangled Representations by Bachem et al.) that Beta-VAE is far from optimal for “extrinsic” disentanglement, the text in section 4.1 should take these results into account. It would also be interesting to contrast with the following paper: Robustly Disentangled Causal Mechanisms: Validating Deep Representations for Interventional Robustness by Bauer et al. which (whilst doing something pretty different) also treats of causality and disentanglement.



**Experience Assessment:**

I have read many papers in this area.

**Review Assessment: Checking Correctness Of Derivations And Theory:**

I assessed the sensibility of the derivations and theory.

**Review Assessment: Checking Correctness Of Experiments:**

I assessed the sensibility of the experiments.

**Review Assessment: Thoroughness In Paper Reading:**

I read the paper at least twice and used my best judgement in assessing the paper.

---

> ### Author Response · Authors · 2019-11-15
> **Reply to Rev#4**
>
> Thank you for your feedback and careful review. We uploaded a revision of the manuscript based on all reviewers’ comments and address yours specifically in this reply.
>
> a.	“It should be made clearer from the start that disentanglement is here only a property of the transformations and that the authors are not trying to define a disentangled internal representation.”
> While we agree and argue in section 2.2 that disentangling transformations is a good starting point, we also think this allows to define a disentangled internal representation. You could actually see our Definition 4 and Proposition 2 as an extension of our framework (initially based on particular transformations) to representations. We clarified this in the discussion below proposition 2 as follows:
> “While we have founded this framework on a functional definition of disentanglement that applies to transformations, the link made here with an intrinsic property of the network allows us to define a disentangled representation as follows: consider of partition of the intermediate representation in several modules, such that their Cartesian product is a factorization of $\mathcal{V}_M$. We can call this partition a disentangled representation because any transformation applied to a given module leads to a valid transformation in the data space (such transformation is relatively disentangled following Def. 9). Interestingly, we obtain that a disentangled representation requires the additional introduction of a partition of the considered set of latent variables into modules. This extra requirement was not considered in classical approaches to disentanglement as it was assumed that each single scalar variables could be considered as an independent module. Our framework provides an insight relevant to artificial and biological systems: as the activity of multiple neurons can be strongly tied together, the concept of representation may not be meaningful at the "atomic" level of single neurons, but require to group them into modules forming a "mesoscopic" level, at which each group can be intervened on independently.”
>
> b.	“how to choose the reference endogenous variables and how to choose the subset E are left entirely to the experiments section. […] section 2 seems precise and formal[...], the procedure to identify modules comes with no guarantees […]”
> We agree more theoretical results would be helpful on that side, they are however challenging to get as the problem we address is novel and hard (the number of partitions of the set of endogenous variables to explore to find putative modules is extremely large). The solution we provide has the benefit to be relatively fast to compute and provides interesting emprical results. We did investigate the effect of several parameters on the quality of clusters (see Fig. 9 in appendix), however this is by far the most computationally expensive aspect of our analysis, especially in (large) high performance generative networks.
>
> In order to address your concern within the time frame of the rebuttal, we introduced at the end of section 3.4. a toy model and an associated theoretical result (proposition 3) to provide insights and justify our NMF based approach. In short, given a simple network with one hidden layer such that each module  has at least one region in the output image only affected by it, we can recover the modules using non-negative matrix factorization of the thresholded influence maps. While this result reflects an ideal case, we think it provides some insights and a starting point to investigate the modular organization of generative neural networks.
>
> c.	“The results presented on CelebA and ImageNet are interesting, in particular using different models is a good idea, however the evaluation relies mostly on a few cases or examples and I would have liked to see more quantitative results, e.g. like in Figure 8 Appendix F. “
> We added more examples in Fig. 12, 13 and 15. By the time of the deadline, we did not have the computational resources to reproduce Fig. 8 appendix F for other models, but we commit to include it in the final version. Additional quantitative assessment is provided in the new Figs. 14 and 16 (see reply to Rev. #3).
>
> d.	“Note on related work: It has been shown [...] that Beta-VAE is far from optimal for “extrinsic” disentanglement, the text in section 4.1 should take these results into account.”
> Many thanks for this suggestion. We updated the beginning of the third paragraph of sec, 4.1. accordingly.
>
> e.	“It would also be interesting to contrast with the following paper: Robustly Disentangled Causal Mechanisms: Validating Deep Representations for Interventional Robustness by Bauer et al. which (whilst doing something pretty different) also treats of causality and disentanglement.”
> Thanks for this suggestion. We added the work of Sutter to the related work section at the end of the introduction section.

---

### Official Review · AnonReviewer1 · 2019-11-07
**Official Blind Review #1**

**Rating:** 8

**Review:**

Summary:
    Authors propose a method to analyze a trained generator (a neural network) from a trained generative model for a distribution of images and produce a family of modules (subsets of neurons) that control certain aspects of the image (upon intervention) such that the different modules identified are "distentangled" in the sense of a lot of work in this area attempting to find latent controllable factors.

This paper has many original ideas and substantial novelty. One of the prime ones is to focus on existing generators and identify subsets of neurons "inside" the generator box that can be manipulated producing meaningful changes in the image. All existing works attempt to constrain the latent space in an appropriate way during training so as to produce different latent factors that manipulate different aspects of the image. Authors point out that disentangled latents cannot be statistically independent. This is very important (and motivates them to group neurons in the model that are statistically dependent) and I cannot agree more with the authors on this point.

Transformation of the image is defined to be intrinsically disentangled if the same transformation can be produced in the image by intervening (transforming) only a small subset of neurons.

Authors assume that the map from z (latents) to image space (g_M) and map from neurons space to image space induced by the generator (\tilde{g}_M) are invertible .

Authors define a class of manipulations of certain subsets of neurons through counterfactuals. Setting the latents to be the same (unique) z that gave rise to the image Y, just intervene on a susbet E of neurons and set them to h (from the valid range of values it can take) - the result of this intervention for this unit (z) produces a counterfactual image corresponding to y. Now, if the counterfactuals corresponding to the manifold of images map back into the manifold, then it is called a faithful counterfactual.

Authors show a very interesting result - Transformation is disentangled if and only if it is produced by a faithful counterfactual mapping. There is an additional result relating modularity of a subset of neurons to disentanglement.

Then authors produce a concrete algorithm inspired by faithful counterfactual mapping as follows. The authors take two images and two latents z_1 and z_2 that produced them. Then they identify a specific neuron under z_1 , then swap that neuron with the value of the neuron from z_2 with everything else remaining the same under z_1. Then they look at the difference in the images and average across all channels. Then they average over all pairs z_1 and z_2 - this representative "image with one channel" is the average effect of manipulating that neuron.

Then, using nonnegative matrix factorization on the thresholded versions of the average effects of all neurons in a layer, they cluster neurons and then they actually form modules. So now each module can be manipulated together when two images are taken together and neurons in the modules are swapped from one into the other.

Pros:
  I really like the key idea of this paper - counterfactually manipulating neurons using its values from that of some other image and observing differences and clustering them to find similar neurons which could be manipulated as a bunch. Some of the results (although seems handpicked) seem pretty good given this is the first work to group neurons inside a pre-trained generator. I have not seen any work on "intrinsic disentanglement" before. This also has additional implications for people interested in GANs. The fact that pre-trained GANs can lead to counterfactually "realistic" images when "concepts" (clustered neurons of this method)  are swapped shows that GANs really do learn something non-trivial about images.

I highly recommend this paper for acceptance. However I would like the authors to address some of my concerns.

Cons:
a)  Typo in Proposition 2 - "then and transformation.." - and should not be there.
b) v_0 is used in some cases - h(z) is used in some places and h (being a constant vector) is used in some places to define counterfactual mapping. Its pretty confusing to read some of the proofs. Is it possible to uniformly define it throughout with constant h and then of course only during experiments and hybridization replace h by v(z_2) (the v from image to be swapped with).
c) Section 3.4 - first paragraph - last line. "Influence maps are grouped by similarity to define modules.." - This is vague. Does this refer to the procedure in the third paragraph of the same section? If so this line could be make to point to this more precise description that comes later.
d) So there are many synthetic experiments where the same pair of images is used to produce many counterfactual images by hybridizing under manipulation of different modules (returned by the clustering algorithm)
What will be useful is for one VAE or GAN ,demonstrate the changes on multiple pairs of images. It seems the image pair is handpicked along with the clustered modules. It would be good to fix the modules and illustrate changes for multiple images using the same set of modules.

e) It seems that the clustered influence maps are simplistic - representing face region, hair region and background. However, ideally we would like to find out - makeup or not, gender , hair color, bangs or not - these are various attributes of the celebA dataset that u get annotated.
Is it possible to correlate obtained modules/influence maps with these annotations already available to find finer concepts that are represented by these annotations ?

One way to find out is - take a module or a neuron - manipulate it by hybridizing and produce a hybrid image from a pair of original images, build a classifier for one of these annotated labels (attributes) on the CelebA dataset .Then do inference on the hybridized image versus the original two images - if the prediction confidence of the attribute classifier changed across original images and also between the hybrid and one of the original- we know that module/neuron is highly correlated with this attribute. One could build a confusion matrix (sort of) between annotated attributes and all elementary (neurons or modules). This could easily tell us if indeed modules/neurons are capturing all attributes we would like.









**Experience Assessment:**

I have read many papers in this area.

**Review Assessment: Checking Correctness Of Derivations And Theory:**

I assessed the sensibility of the derivations and theory.

**Review Assessment: Checking Correctness Of Experiments:**

I carefully checked the experiments.

**Review Assessment: Thoroughness In Paper Reading:**

I read the paper at least twice and used my best judgement in assessing the paper.

---

> ### Author Response · Authors · 2019-11-15
> **Reply to Rev#1**
>
> Thank you for the detailed and positive review of our work. We uploaded a revision of the manuscript based on all reviewers’ comments and address yours specifically in this reply.
>
> a) “Typo in Proposition 2 - "then and transformation.." - and should not be there.“
> Apologies for this typo that we noticed after submission, we meant “then any transformation”. This has been fixed in the revised version.
>
> b) “v_0 is used in some cases - h(z) is used in some places and h (being a constant vector) is used in some places to define counterfactual mapping. […]”
> Many thanks for noticing this inconsistency originating from an earlier version of the result. We updated Def. 7 and proof of Prop. 1 accordingly.
>
> c) ““Section 3.4 - first paragraph - last line. "Influence maps are grouped by similarity to define modules.." - This is vague. […] this line could be make to point to this more precise description that comes later."
> Indeed we describe this in the third paragraph of this section. We added “, as we will describe in detail below” at the end of this sentence to clarify.
>
> d) “It would be good to fix the modules and illustrate changes for multiple images using the same set of modules. “
> In the novel Figures 12, 13 and 15 (in appendix), we now provide a large number of hybrids for the BEGAN trained on CelebA and the BIGGAN trained on ImageNet. We picked the pairs based on the quality of the original samples (before hybridization) and in order to have some variations in their properties within and across pairs. For a given layer and each sample, all clustered modules are intervened on to provide all hybrids.
>
> e) “It seems that the clustered influence maps are simplistic […]. However, ideally we would like to find out - makeup or not, gender , hair color, bangs or not [...]. Is it possible to correlate obtained modules/influence maps with these annotations already available to find finer concepts that are represented by these annotations ? One way to find out is - take a module or a neuron - manipulate it by hybridizing and produce a hybrid image from a pair of original images, build a classifier for one of these annotated labels (attributes) on the CelebA dataset .[…]”
> Finding more refined clusters is definitively an important research direction, perhaps limited by the fully unsupervised approach we take here: clustering algorithms typically capture dominant variations in the data, but are challenged when we want to increase the number of clusters. Also given disentanglement of specific properties was not enforced at training (and especially not supervised), it is an open question how far subtle properties defined from a human perspective (make up, etc…) will correspond to specific modules enforced by the learning algorithm and the architecture. We will definitely work on dedicated approaches in this direction but leave it to a future study.
>
> While the training of classifiers dedicated to CelebA features required too much work and computational time in the context of this revision, we think what you describe is to some extent akin to what our use of classifiers in the BIGGAN experiment on the ImageNet dataset (Figs. 5 and 17, Table 3). Indeed, Fig. 5 depicts how the counterfactuals affect the decision of the classifier regarding which object is in the image. In addition, in this revision, we also use a classifier to assess the quality of the hybrids based on the entropy of the classifier's probabilistic decision (see Fig. 14 and 16, and reply to Rev. #3). While we think these result are more novel and challenging than what we could do on CelebA, we suggest to perform (upon request) gender classification and study the impact of counterfactuals on it in the final version.
>
> Your comment also puts forward that we could use classifiers to find the modules. This relates to work in the literature (see e.g. Bau et al, 2018, that we mention in related work). In contrast, the present paper is focused on an unsupervised approach, aiming at extracting modules from non-annotated data (dataset annotation relying on heavy and tedious human work).

---

### Decision · Program_Chairs · 2019-12-19

**Decision:**

Accept (Poster)

**Comment:**

This paper provides a fresh application of tools from causality theory to investigate modularity and disentanglement in learned deep generative models. It also goes one step further towards making these models more transparent by studying their internal components. While there is still margin for improving the experiments, I believe this paper is a timely contribution to the ICLR/ML community.
This paper has high-variance in the reviewer scores. But I believe the authors did a good job with the revision and rebuttal. I recommend acceptance.